# Lactate sensing mechanisms in arterial chemoreceptor cells

Hortensia Torres-Torrelo [1,2,4,5], Patricia Ortega-Sáenz[1,2,3,5], Lin Gao [1,2,3] & José López-Barneo [1,2,3]✉

Classically considered a by-product of anaerobic metabolism, lactate is now viewed as a fundamental fuel for oxidative phosphorylation in mitochondria, and preferred over glucose by many tissues. Lactate is also a signaling molecule of increasing medical relevance. Lactate levels in the blood can increase in both normal and pathophysiological conditions (e.g., hypoxia, physical exercise, or sepsis), however the manner by which these changes are sensed and induce adaptive responses is unknown. Here we show that the carotid body (CB) is essential for lactate homeostasis and that CB glomus cells, the main oxygen sensing arterial chemoreceptors, are also lactate sensors. Lactate is transported into glomus cells, leading to a rapid increase in the cytosolic NADH/NAD$^+$ ratio. This in turn activates membrane cation channels, leading to cell depolarization, action potential firing, and Ca$^{2+}$ influx. Lactate also decreases intracellular pH and increases mitochondrial reactive oxygen species production, which further activates glomus cells. Lactate and hypoxia, although sensed by separate mechanisms, share the same final signaling pathway and jointly activate glomus cells to potentiate compensatory cardiorespiratory reflexes.

[1] Instituto de Biomedicina de Sevilla (IBiS), Hospital Universitario Virgen del Rocío/CSIC/Universidad de Sevilla, Seville, Spain. [2] Departamento de Fisiología Médica y Biofísica, Facultad de Medicina, Universidad de Sevilla, Seville, Spain. [3] Centro de Investigación Biomédica en Red sobre Enfermedades Neurodegenerativas (CIBERNED), Madrid, Spain. [4] Present address: Facultad de Medicina, Universidad San Pablo-CEU, CEU Universities, Madrid, Spain. [5] These authors contributed equally: Hortensia Torres-Torrelo, Patricia Ortega-Sáenz. ✉email: lbarneo@us.es

While lactate has classically been considered a waste product of anaerobic metabolism and a fatigue agent, this view has changed radically in recent years. It is now established that even under fully aerobic conditions, lactate is the final product of glycolysis in many cell types[1,2]. Lactate can be secreted, for example, by white skeletal muscle, erythrocytes or astrocytes and either consumed by neighboring cells (e.g., neurons)[3] or transported by the blood to cells in distant organs for energy production (e.g., in the heart) or gluconeogenesis (e.g., in the liver)[4]. In lactate-consuming cells, lactate is converted to pyruvate by lactate dehydrogenase in the cytosol or in the mitochondrial intermembrane space[5–8]. In addition to being a systemic metabolic substrate, preferred over glucose as a fuel in numerous tissues[9], lactate is gaining relevance for its role as a signaling molecule. Lactate is co-transported with protons ($H^+$) into cells by a broadly distributed family of reversible monocarboxylate transporters (MCTs)[10]. It can also bind to GPR81, a G-protein coupled putative lactate receptor expressed in adipose tissue and brain[11,12]. Lactate inhibits lipolysis[13], favors angiogenesis through the induction of VEGF[14] and elicits vasodilation[15]. Infused lactate increases circulating BDNF levels[16], protects heart and ischemic neurons, promotes adult hippocampal neurogenesis[17,18], and inhibits inflammation following organ injury[19,20]. Moreover, lactate's interaction with histidine residues activates TREK1 $K^+$ channels[21], while lactylation of histone lysines regulates gene expression[22].

Despite the vast amount of existing literature on lactate metabolism and signaling in numerous organs, the "where" and "how" blood lactate (lactatemia) is sensed to elicit adaptive responses in the organism are unknown. Acute hyperlactatemia occurs in response to physical exercise or exposure to systemic hypoxia[4,7,23], however, the homeostatic role of these responses is not well understood. It has been suggested that lactate itself, apart from lactic acidosis, can induce ventilation[24] and contribute to the initiation and maintenance of the hypoxic ventilatory response (HVR)[25], possibly via an action on the carotid body (CB)[26], the main acute $O_2$-sensing organ in mammals[27]. It has also been reported that lactate activates the CB by means of binding to an atypical olfactory receptor (Olfr78) that is highly expressed in the CB's chemoreceptor (or glomus) cells[28]. This latter proposal was disregarded because the findings could not be reproduced by others[29–32]. Nonetheless, in preliminary studies, we found that single Olfr78-deficient glomus cells are indeed directly activated by lactate[31]. Here we report that the CB is essential for lactate homeostasis and that glomus cells are physiologically relevant acute lactate sensors. Lactate is transported into glomus cells where it increases the cytosolic NADH/NAD$^+$ ratio, decreases cytosolic pH, and stimulates mitochondrial production of reactive oxygen species (ROS), which in turn activate cation channels to induce membrane depolarization, $Ca^{2+}$ entry, and transmitter release. Lactate and hypoxia, although sensed by separate mechanisms, jointly activate glomus cells to potentiate adaptive cardiorespiratory reflexes.

## Results and discussion

**Carotid body dysfunction enhances hypoxia-induced lactatemia.** The main systemic acute $O_2$-sensing organ in mammals is the CB, which upon activation by hypoxia stimulates the central respiratory and autonomic centers to induce hyperventilation and sympathetic activation[27]. We tested whether the CB plays any role in lactate homeostasis by monitoring hypoxia-induced lactatemia in wild type mice (control) compared to mice insensitive to hypoxia due to deficient CB function. To this end, we used two mouse models previously studied in our laboratory: TH-HIF2α, with embryonic ablation of the gene *Epas1* (coding HIF2α) in

sympathoadrenal (tyrosine hydroxylase-positive) cells and exhibiting CB atrophy and strong inhibition of the HVR[33] (Supplementary Fig. 1), and ERT2-HIF2α, with ubiquitous conditional ablation of *Epas1* and showing normal CB development but selective abolition of responsiveness to hypoxia in adulthood[34]. In wild type mice, hypoxia, even if relatively mild, induced a graded and reversible increase in blood lactate within a few minutes that occurred in parallel with an increase in ventilation (Fig. 1a and c). In contrast, hypercapnia (5% $CO_2$), a stimulus known to activate CB cells[27], had no effect on blood lactate but did evoke a ventilatory response stronger than that induced by hypoxia (Fig. 1b and c). Lactatemia elicited by breathing 10% $O_2$, was much higher in CB-deficient mice (~30% and ~90% increase in TH-HIF2α and ERT2-HIF2α strains, respectively) than in controls (Fig. 1d). These data indicate that although hypercapnia induces, as hypoxia, CB activation[27] and hyperventilation it is not accompanied by lactatemia. CB activation during systemic hypoxia in normal mice reduces the intensity of hypoxemia[33] and thereby blunts lactate release from tissues.

**$Ca^{2+}$-dependent activation of carotid body glomus cells by lactate.** We studied the effect of extracellular L-lactate (sodium salt) on mouse single glomus cells, the polymodal acute $O_2$ sensing neurosensory elements in the CB (Fig. 2a)[27]. Amperometric experiments showed that lactate induced a dose-dependent exocytotic catecholamine release from cells in CB slices (Fig. 2b and c). Cytosolic $Ca^{2+}$ measured in Fura 2-loaded dispersed glomus cells increased in response to high $K^+$ and hypoxia[35]. In ~90% of these $O_2$-sensitive cells, lactate also elicited an increase in cytosolic $Ca^{2+}$ (Fig. 2d–f) which was abolished by extracellular cadmium ($Cd^{2+}$), a non-selective blocker of high-threshold membrane $Ca^{2+}$ channels in glomus cells (Fig. 2g)[35]. Lactate-induced secretory activity was also blocked by $Cd^{2+}$ (Fig. 2h) and strongly inhibited by nifedipine (Fig. 2i and j), a blocker of L-type $Ca^{2+}$ channels in rodent glomus cells[36]. Therefore, similar to hypoxia, lactate activates glomus cells in an external $Ca^{2+}$-dependent manner. In agreement with these data in mouse CB, we recorded robust responses to lactate in dispersed rat glomus cells (Supplementary Fig. 2a) as well as in cells in rat CB slices (Supplementary Fig. 2b). It has recently been reported that membrane potential and conductance of rat glomus cells are unaffected by lactate, however, these experiments were performed with the whole-cell configuration of the patch-clamp technique and therefore intracellular dialysis may have altered the lactate signaling pathway[37] (see below).

**Lactate is transported into glomus cells and leads to NADH accumulation and mitochondrial ROS production.** In many cell types, lactate is co-transported with $H^+$ across the plasma membrane by monocarboxylate transporters (MCTs) and once inside the cell is converted to pyruvate (in the cytosol or organelles) by lactate dehydrogenase with production of NADH (see scheme in Fig. 3a). We studied by immunocytochemistry the expression in CB cells of MCT1, 2, and 4, the most representative MCTs in mammalian cells[38]. MCT2, a high-affinity (Km < 1 mM) lactate transporter expressed in neurons and other cell types[38,39], was consistently detected in tyrosine hydroxylase (TH) positive glomus cells (Fig. 3b), however, this transporter was absent in glia-like, fibrillary acidic protein (GFAP) positive, CB type II cells (Fig. 3c; Supplementary Fig. 3). MCT2 was also expressed in neurons of the superior cervical ganglion (SCG) and in adrenal medulla (AM) chromaffin cells (Supplementary Fig. 4a and b), which are catecholaminergic (TH positive) cells that, as the CB glomus cells, derive from neural crest progenitors. Neither CB glomus cells (Fig. 3d) nor type II cells (Supplementary Fig. 5a and c) expressed MCT1, a ubiquitous lactate transporter with lower affinity (Km ~5 mM) than MCT2[38].

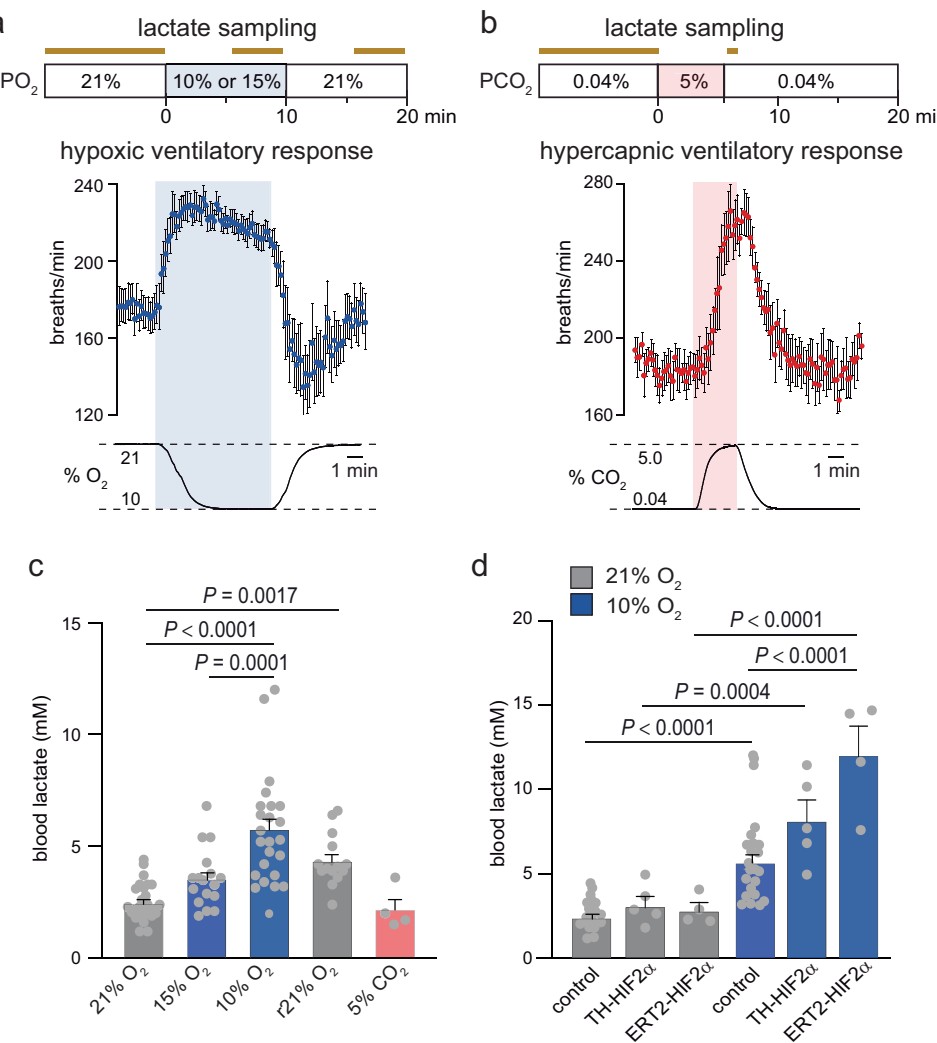

**Fig. 1 Blunting of hypoxia-induced lactatemia by arterial chemoreceptors. a, b** Top. Scheme of the experimental protocol followed to measure blood lactate (lactate sampling indicated by horizontal brown bars) under normoxic conditions (21% $O_2$) and during exposure to hypoxia (10% or 15% $O_2$ tension, pale blue) or hypercapnia (5% $CO_2$, pale red). Bottom. Plethysmographic recordings showing the average increase in respiratory frequency (breaths/min) during exposure to hypoxia (blue dots; $n = 35$ mice) or hypercapnia (red dots; $n = 27$ mice). **c** Scatter plots of blood lactate levels measured in the various experimental conditions. Values (mean ± SEM) are: normoxia (2.5 ± 0.1 mM, $n = 29$ mice; gray), hypoxia 15% $O_2$ (3.6 ± 0.3 mM, $n = 17$ mice; blue), hypoxia 10% $O_2$ (5.8 ± 0.5 mM, $n = 23$ mice; blue), recovery in normoxia (r21% $O_2$ measured 5 min after returning to normoxia; 4.4 ± 0.3 mM, $n = 14$ mice; gray), and hypercapnia (2.2 ± 0.5 mM, $n = 4$ mice; pale red). P-values calculated by one-way ANOVA followed by Tukey's multiple comparisons post hoc test are indicated. **d** Scatter plots of blood lactate levels, during normoxia (gray) and hypoxia (blue), in control and mice with carotid body dysfunction due to embryonic or conditional adult deficiency of Hif2α. P-values calculated by one-way ANOVA followed by Tukey's multiple comparisons post hoc test are indicated. Mean ± SEM values are: TH-HIF2α (normoxia = 3.2 ± 0.6 mM; hypoxia 10% $O_2$ = 8.3 ± 1.2 mM, $n = 5$ mice); ERT2-HIF2α (normoxia 2.9 ± 0.4 mM; hypoxia 10% $O_2$ = 12.3 ± 1.7 mM, $n = 4$ mice). Source data are provided as a Source data file.

However, TH positive CB cells expressed MCT4 (Fig. 3e), which was considered to be a low affinity (Km > 20 mM) transporter characteristic of lactate secreting cells[38,40]. A recent study indicates that MCT4 affinity for lactate (Km ~1 mM) is much higher than previously thought and that this transporter has a relevant affinity for pyruvate (Km ~4 mM). However, the higher affinity for lactate relative to pyruvate makes MCT4 suitable for a major role in lactate release from cells[41]. MCT4 appeared to be only slightly expressed in some type II cells (Supplementary Fig. 5b and d). The selective abundant expression of MCT2 in glomus cells suggest that they can easily uptake lactate from blood, a property compatible with their role as lactate sensors. MCT4 may also contribute to lactate uptake by glomus cells. This transporter, which is less sensitive to pyruvate than to lactate[38,41], could also mediate lactate secretion from glomus cells in some circumstances, as for example in conditions of

accelerated glycolysis secondary to mitochondrial dysfunction[42]. MCT4 might also be preferentially expressed in CB neuroblasts, which are immature TH-positive CB cells with the mitochondria-based acute $O_2$-sensor still undeveloped[43]. Noteworthy, type II cells, which are glia-like supportive elements with quiescent stem cell function[44,45], did not express MCT1 and seemed to have only low levels of MCT4 expression. In this regard, type II cells are different from astrocytes which release lactate to fuel and modulate the function of neighboring neurons[3,39,46]. In agreement with the immunocytochemical observations, we monitored lactate transport into glomus cells by measuring intracellular acidification using single-cell microfluorimetry (Fig. 3f and g). Control of the lactate-induced acidification signal was obtained by exposure of the cells to $CO_2$, which strongly acidifies the cytosol due to its conversion to carbonic acid by carbonic anhydrase present in CB glomus cells[27].

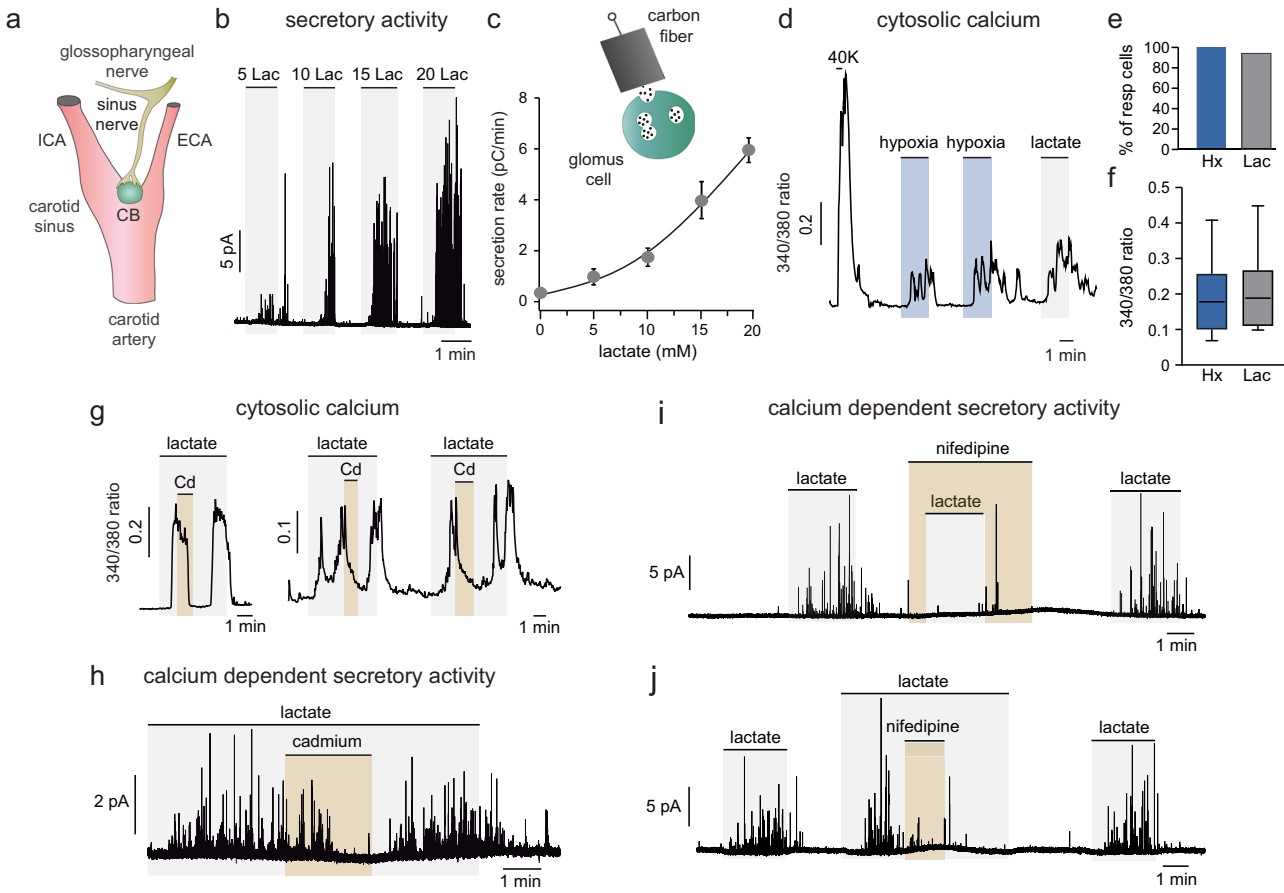

**Fig. 2 Activation of carotid body cells by lactate. a** Scheme of the carotid bifurcation representing the location of the carotid body (CB) and sensory innervation (ICA and ECA, internal and external carotid artery). **b**, **c** Secretory activity (catecholamine release measured with a polarized carbon fiber) induced by extracellular lactate in glomus cells in CB slices. Average values (mean ± SEM) in (**c**) are from: 5 Lac ($n = 5$ cells/5 mice); 10 Lac ($n = 8$ cells/8 mice); 15 Lac (6 cells/6 mice); 20 Lac ($n = 5$ cells/5 mice). **d** Increase in cytosolic $Ca^{2+}$ induced by $K^+$ (40 mM), hypoxia ($O_2$ tension $\approx 15$ mm Hg, blue), and lactate (5 mM, gray) in dispersed single glomus cells. **e** Percentage of cells recorded that responded to hypoxia (blue) and lactate (5–10 mM; gray) ($n = 20$ cells from 6 mice). **f** Box plots summarizing the amplitude of the responses to hypoxia and lactate ($n = 20$ cells from 6 mice). Mean ± SEM values are: hypoxia: 0.19 ± 0.02; 5 mM lactate: 0.21 ± 0.02. In the boxplots the middle line is the median, the lower and upper hinges (IQR, interquartile range) correspond to the first and third quartiles, the upper whisker extends from the hinge to the largest value no further than 1.5 × IQR and the lower whisker extends from the hinge to the smallest value at most 1.5 × IQR. **g** Representative recordings ($n = 5$) of lactate (5 mM)-induced rise in cytosolic $Ca^{2+}$ and its inhibition by $Cd^{2+}$. **h** Representative amperometric recording illustrating the reversible inhibition of lactate (10 mM)-induced secretory activity by extracellular $Cd^{2+}$. Similar recordings were obtained in three separate experiments. **i**, **j** Amperometric recordings of the inhibitory effects of nifedipine (0.5 µM) on lactate (10 mM)-induced secretion applied before and after exposure to lactate (4 experiments in 3 mice). Source data are provided as a Source data file.

Exposure of dispersed glomus cells to lactate elicited a fast, highly reversible, and dose-dependent increase in intracellular NADH, as monitored by single-cell NAD(P)H autofluorescence[34,42,47,48] (Fig. 4a and b). This signal was strongly inhibited by AR-C155858 (Fig. 4c and d), an inhibitor of MCT1 and MCT2[49]. The lactate (10 mM)-dependent increase in NADH was abolished by the extracellular application of pyruvate (5 mM), a molecule also co-transported with $H^+$ by MCT1 and MCT2 but with a higher affinity (Km < 1 mM and <0.1 mM for MCT1 and MCT2, respectively) than lactate[38,39]. Application of a short pulse of pyruvate produced a marked decrease in the lactate-induced NADH signal to values below that even to basal levels, indicating that once inside the glomus cell, pyruvate, by consuming NADH, was rapidly converted to lactate by lactate dehydrogenase (Fig. 4e and f). Treatment with α-ketobutyrate (α-KB), which was shown previously to rapidly decrease NADH autofluorescence in glomus cells due to its conversion to non-metabolizable α-hydroxybutyrate[50], also prevented the lactate-dependent NADH rise in the cells (Supplementary Fig. 6a and b).

In addition to decreasing NADH levels, pyruvate inhibited the secretory response to lactate in glomus cells (Fig. 4g). Even a small amount of extracellular pyruvate (100 µM), near the normal concentration in plasma[51], was sufficient to partially inhibit lactate-induced secretion in glomus cells (Supplementary Fig. 6c and d). However, normal responses to high lactate were recorded in cells bathed by solutions containing physiological plasma values of pyruvate and lactate[51] (Fig. 4h). Although in the presence of lactate a short exposure to pyruvate consistently inhibited the secretory activity in glomus cells, in most cases this inhibitory phase was followed by a late activation of the cells (Supplementary Fig. 6e). Indeed, although at high non-physiological concentrations, pyruvate by itself (in the absence of lactate) induced a slow secretory activity in all cells tested probably due to intracellular acidification (Supplementary Fig. 6f). These results suggest that increases in extracellular lactate alter the extra and intracellular pyruvate/lactate equilibria, resulting in NADH accumulation (or an increase in the $NADH/NAD^+$ ratio), which participates in the regulation of $Ca^{2+}$ influx by lactate in

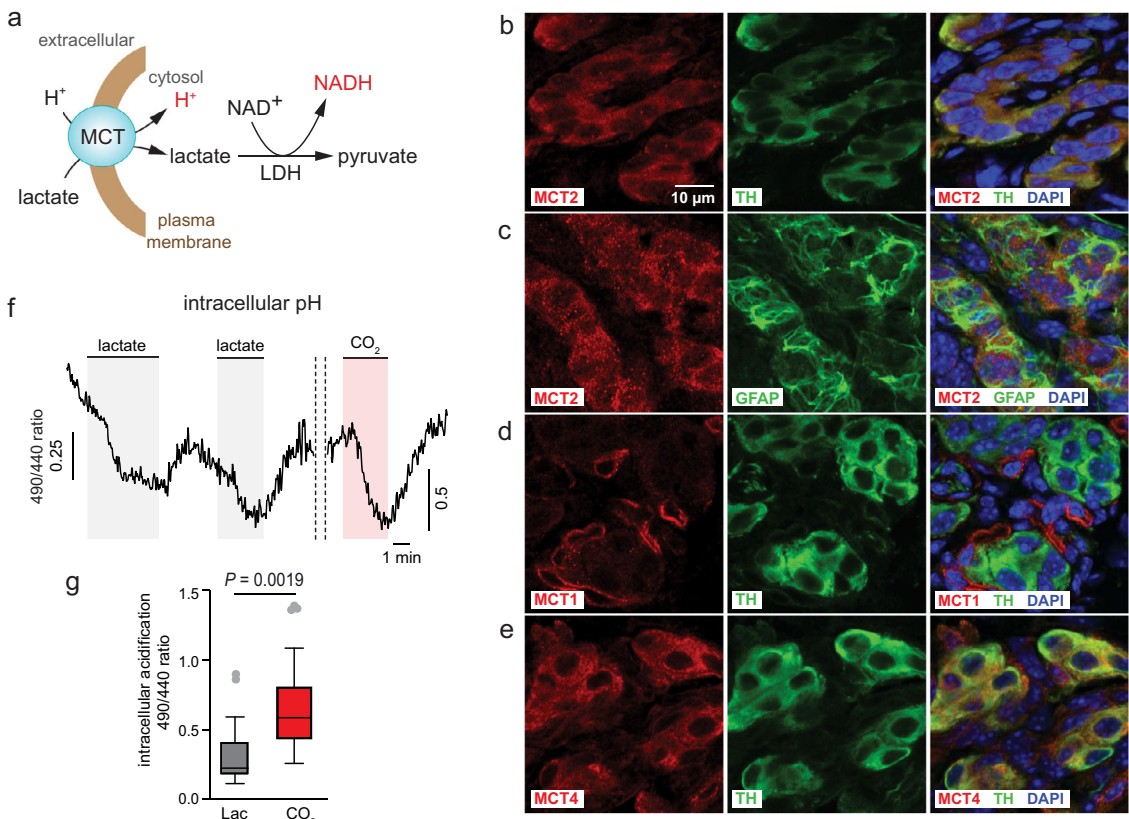

**Fig. 3 Expression of lactate transporters in glomus cells. a** Scheme of proton-coupled lactate co-transport across the plasma membrane and conversion of lactate to pyruvate by lactate dehydrogenase (LDH). MCT (monocarboxylate transporter). **b**, **c** Immunocytochemical demonstration of MCT2 expression in TH-positive CB glomus cells and absence of MCT2 expression in GFAP-positive type II cells. **d** Lack of expression of MCT1 in TH-positive CB glomus cells. **e** Expression of MCT4 in TH-positive CB glomus cells. Photographs in (**b**–**e**) are representative examples of similar data obtained in $n = 4$ mice. Scale bar in (**b**, left) is the same for all panels. **f** Representative microfluorimetric recording of intracellular glomus cell acidification induced by lactate (Lac, 10 mM) and hypercapnia (~12% $CO_2$). **g** Box plots representing the distribution of the amplitude of pH signals induced by the two stimuli ($n = 16$ cells from 2 mice). Mean ± SEM values are: 10 mM lactate (gray): 0.36 ± 0.6; 12% $CO_2$ (red):0.83 ± 0.13. The boxplots indicate median (middle line), 25th, 75th percentile (box), and largest and smallest values extending no further than 1.5× interquartile range (whiskers). Data beyond the end of the whiskers (outlying points) are plotted individually (gray). The $P$-value calculated by two tails, paired $t$ test are indicated. Source data are provided as a Source data file.

these cells (see below). In addition, intracellular acidification produced by lactate transport may also contribute to activation of glomus cells. Together with NADH measurements we monitored the generation of mitochondrial ROS in glomus cells using a fluorescent genetic probe selectively directed to the mitochondrial matrix or intermembrane space (IMS)[50]. Lactate produced a dose-dependent reversible rise in matrix ROS (Fig. 4i, top), which likely reflected the increased fuel supply to the highly active glomus cell mitochondria[50,52]. Indeed, pyruvate at the same concentration also produced an increase in matrix ROS, which was in contrast to the decrease in matrix ROS measured in response to hypoxia (Fig. 4i, bottom and j)[50]. The increase in mitochondria ROS produced by lactate or pyruvate was also manifested at the IMS (Supplementary Fig. 6g and h).

**Lactate-induced depolarization and action potential firing is triggered by NADH-dependent cation channels.** Because lactate induction of secretory activity in glomus cells was dependent on $Ca^{2+}$ influx, we hypothesized that lactate might elicit membrane depolarization and, in this way, give rise to the opening of voltage-gated $Ca^{2+}$ channels. Experiments on patch-clamped (perforated patch configuration) dispersed glomus cells showed that lactate (10 mM) elicited a reversible depolarization of 9.5 ± 1 mV ($n = 17$ cells) from a relatively low spontaneous basal

resting potential (RP ≈ −30 to −40 mV) (Fig. 5a and b). When current clamped cells were hyperpolarized to ~ −70 mV to de-inactivate voltage-gated ion channels, lactate evoked a larger (31 ± 4 mV; $n = 16$) depolarization accompanied by the firing of action potentials (Fig. 5b and c). Because glomus cells have a $Na^+$-permeant standing conductance at rest[53,54], replacement of extracellular $Na^+$ with N-methy-D-glucamine (NMDG, a large membrane-impermeant cation) resulted in cell hyperpolarization and abolition of the response to lactate (Fig. 5d). As shown above for the recordings of NADH autofluorescence and glomus cell secretion, application of a fast pulse of pyruvate rapidly and reversibly inhibited the cell depolarization and action potential firing induced by lactate (Fig. 5e). Finally, the electrophysiological effects of lactate (Fig. 5f) and lactate-induced secretory activity (Fig. 5g) were abolished by 2-APB, a non-selective blocker of some transient receptor potential (TRP) and other cationic channels in CB[55]. Taken together, these data suggest that lactate-induced increase in the cytosolic NADH/NAD+ ratio leads to a cation channel-dependent depolarization of glomus cells with subsequent opening of $Ca^{2+}$ channels, $Ca^{2+}$ influx, and trans-mitter release. SCG neurons and AM chromaffin cells showed lactate-induced increases in NAD(P)H autofluorescence (Sup-plementary Fig. 7a, c, e), which is compatible with the expression of MCT2 in these cells (see Supplementary Fig. 4). However,

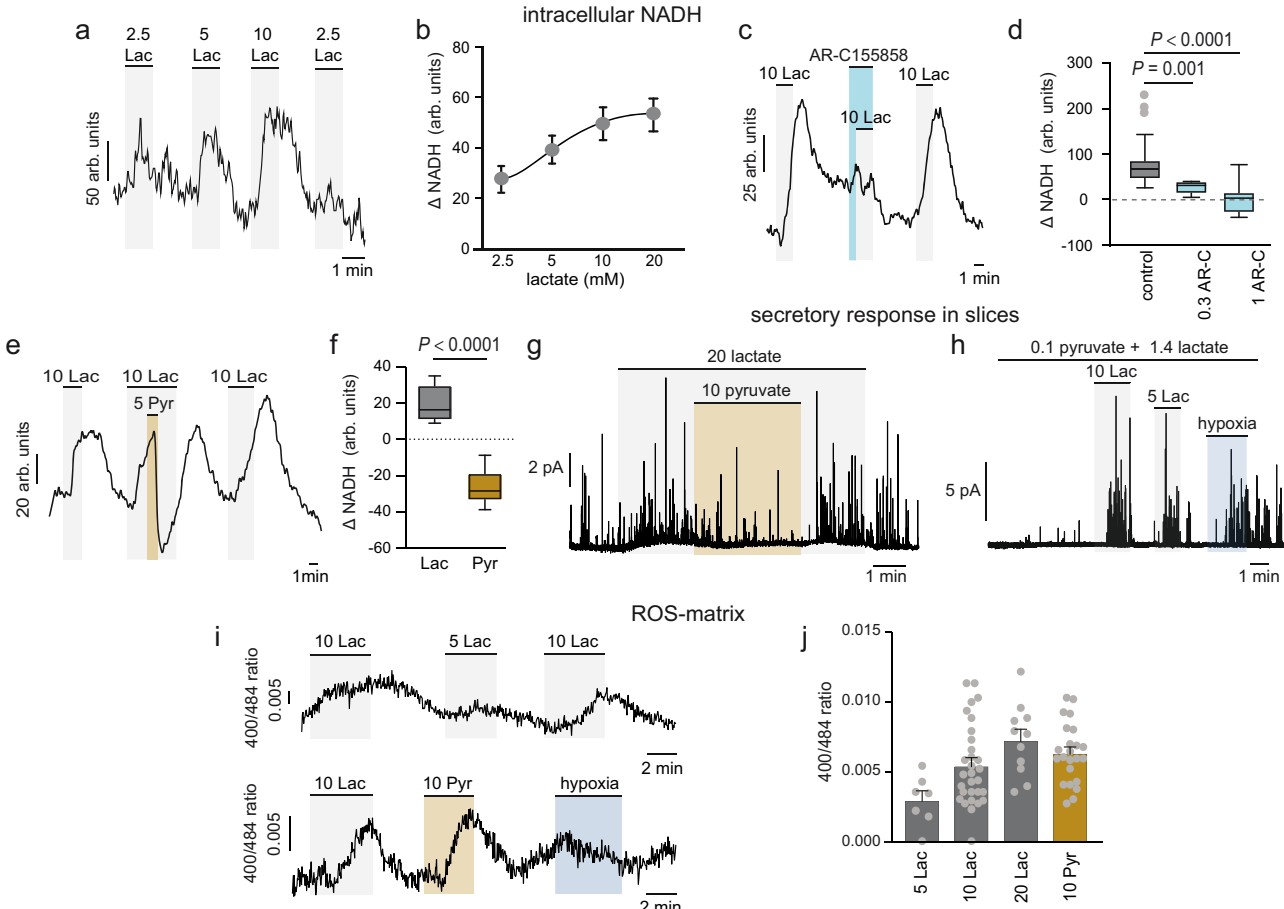

**Fig. 4 Lactate-induced intracellular NADH and ROS signals. a, b** Increase in NAD(P)H autofluorescence (ΔNADH) induced by lactate (Lac in mM, gray) in dispersed glomus cells. Average values (mean ± SEM) in (**b**) are from: 2.5 Lac ($n = 21$ cells/6 mice); 5 Lac ($n = 27$ cells/8 mice); 10 Lac (28 cells/7 mice); 20 Lac ($n = 25$ cells/4 mice). **c** Reversible blockade of a lactate (10 mM, gray)-induced rise in intracellular NADH (ΔNADH) by MCT inhibition with AR-C155858 (0.3 μM, light blue). **d** Box plots representing the distribution of lactate-induced NADH signals under control conditions (gray) and with two different doses (0.3 and 1μM; light blue) of AR-C155858 ($n = 22$ cells from 4 mice). Statistical non-parametric tests were used. Calculated mean ± SEM values are: 10 mM lactate: 87.9 ± 11.7; 0.3 μM of AR-C155858: 28.8 ± 6.4; 1 μM of AR-C155858: 3.2 ± 8.5. $P$ values calculated with Mann–Whitney test are indicated. **e** Inhibition of lactate (10 mM, gray)-induced NADH signal by the application of a short pulse of pyruvate (Pyr, 5 mM, light brown). **f** Box plots representing the distribution of changes in NADH signals (ΔNADH) induced by lactate (10–20 mM, gray) and following exposure to pyruvate (5–10 mM, light brown) ($n = 10$ cells from 5 mice). Calculated mean ± SEM values are: 10–20 mM lactate: 22.8 ± 4.0; 5–10 mM pyruvate: −32.3 ± 6.1. The boxplots in (**d, f**) indicate median (middle line), 25th, 75th percentile (box), and largest and smallest values extending no further than 1.5× interquartile range (whiskers). Data beyond the end of the whiskers (outliers) are plotted individually (gray). The $P$ value calculated with Mann–Whitney test is indicated. **g** Representative amperometric recordings illustrating the inhibition of lactate (20 mM, gray)-induced secretion in a glomus cell by pyruvate (10 mM, light brown) (similar results were obtained in $n = 7$ cells from 5 mice). **h** Representative amperometric recordings illustrating the secretory response to high (5 and 10 mM) lactate (gray) and hypoxia (≈15 mm Hg; pale blue) of glomus cells maintained in physiological resting levels of pyruvate/lactate concentrations (0.1 and 1.4 mM, respectively). **i** Reversible and dose-dependent increase in mitochondrial matrix ROS induced by lactate (top and bottom; gray) and pyruvate (bottom; light brown) at the indicated concentrations (mM). Note that hypoxia ($O_2$ tension ≈ 15 mm Hg, pale blue) induces a decrease in matrix ROS. **j** Average (mean ± SEM) increases in matrix ROS induced by lactate or pyruvate (in mM). Data are from: 2.5 Lac ($n = 23$ cells/3 mice); 5 Lac ($n = 7$ cells/2 mice); 10 Lac ($n = 30$ cells/3 mice); 20 Lac ($n = 11$ cells/2 mice). Arbitrary units (arb. units). Source data are provided as a Source data file.

patch clamped dispersed SCG neurons that were reversibly depolarized by high extracellular K⁺ were unaltered, or even slightly hyperpolarized, by lactate (Supplementary Fig. 7b, d). On the other hand, hypercapnia induced catecholamine release from adult AM chromaffin cells[56], however, these cells appeared to be practically insensitive to lactate (Supplementary Fig. 7f). These observations further support a selective role as lactate sensors for CB glomus cells. The differential sensitivity of glomus cells to lactate, respecting SCG neurons or AM chromaffin cells, may be a consequence of the different expression of membrane ion channels. However, it could be also due to the fact that neither SCG

neurons nor AM chromaffin cells are able to generate a relevant mitochondrial ROS signal in response to lactate.

Pyridine nucleotides are promiscuous signaling molecules that can regulate, directly or indirectly, a large variety of ion channel families[57], most of which are represented in glomus cells[27,30,52,58] and therefore could mediate lactate-dependent depolarization. A good candidate forming part of this response is the TRPC5 channel because the mRNA coding this protein is highly abundant in glomus cells[30] and in a comparative microarray analysis we showed that TRPC5 mRNA is more highly expressed in the CB than in SCG or AM cells[52]. To test this possibility, we performed experiments on CB glomus cells from TRPC5

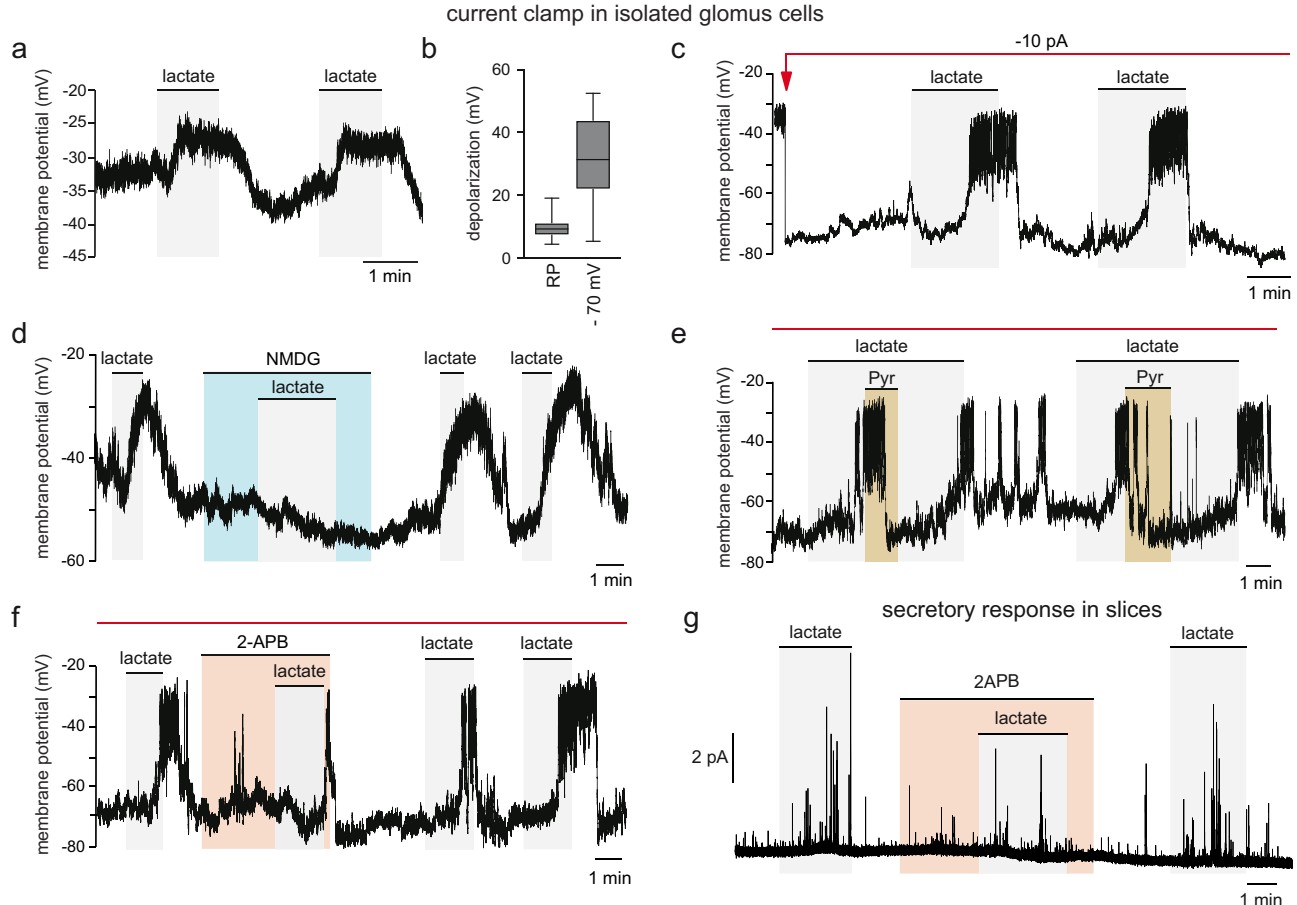

**Fig. 5 Depolarization and action potential firing induced by lactate in glomus cells. a–c** Representative examples of lactate (10 mM)-induced reversible depolarization in a current clamped cell (perforated patch) maintained at the spontaneous resting potential (RP ≈ 30-40 mV). **c** In cells hyperpolarized to ~ −70 mV by current injection (red line and arrow) lactate (10 mM) induced larger depolarizations and the firing of action potentials. **b** Box plots representing the amplitudes of lactate (10 mM)-induced depolarizations in the two conditions. Mean ± SEM values were: 9.5 ± 1 mV ($n = 17$ cells from 16 mice) at RP; 31.2 ± 4 mV ($n = 16$ cells from 10 mice) at ~ −70 mV. The boxplot represents the median (middle line), 25th, 75th percentile (box), and largest and smallest values extending no further than 1.5× interquartile range (whiskers). Source data are provided as a Source data file. **d** Cell hyperpolarization and abolition of the lactate (10 mM)-induced depolarization after replacement of extracellular $Na^+$ with N-methyl-D-glucamine (NMDG). Similar results were obtained in all cells tested (7 cells from 4 mice). **e** Inhibition of lactate (10 mM)-induced depolarization and action potential firing by application of a short pulse of pyruvate (Pyr, 5 mM). Representative example of 8 cells recorded from 5 mice. Red line indicates current injection through the recording electrode to maintain a resting potential of ~ −70 mV. **f** Inhibition of lactate (10 mM)-induced depolarization and action potential firing by 2-APB (10 μM), a non-selective cation channel blocker. Representative example of 3 cells recorded from 3 mice. Red line indicates current injection through the recording electrode to maintain a resting potential of approximately −70 mV. Note that action potential amplitude is truncated due to the relatively long sampling interval (4 ms). **g** Reversible inhibition of the lactate (20 mM)-induced secretory response by 2-APB (10 μM). Representative example of 4 experiments in 3 mice.

knockout mice, which exhibited normal responses to hypoxia and lactate (Supplementary Fig. 8a–c). We also studied CB glomus cells from mice lacking TRPC6 channels, another member of the canonical sub-family of TRP channels[59], which is expressed in glomus cells[58] and has been suggested to be essential for hypoxic pulmonary vasoconstriction[60]. Our data indicated that sensitivity to lactate was not significantly affected in either TRPC6-deficient glomus cells or in cells obtained from *Trpc5/Trpc6* double knockout mice (Supplementary Fig. 8d–g). In addition to TRPC5, mRNAs coding for subunits of TRPC3 and TRPM7 channels, as well as a beta subunit of non-voltage sensitive epithelial $Na^+$ channels have been reported to be abundant in glomus cells[30]. Nonetheless, pharmacological experiments with several specific blockers (Pry3, FTY720, and amiloride for TRPC3, TRPM7, and epithelial $Na^+$ channels, respectively)[61–63] showed that these channel types are not essential for lactate sensing by glomus cells (Supplementary Fig. 9a–f). These results suggest that the

intracellular NADH and ROS produced by lactate are promiscuous signals that probably redundantly modulate several types of cationic channels in glomus cells.

**Lactate- and O₂-sensing are mediated by separate mechanisms**. It is well established that activation of CB glomus cells by hypoxia is a cell autonomous process that can be elicited in the absence of lactate or any other endocrine or paracrine signal[27]. Recently, it was shown that hypoxia-induced glomus cell transmitter release depends on both NADH, shuttled from the mitochondria to the cytosol, and ROS that are primarily generated at the mitochondrial IMS (probably at mitochondrial complexes—MC-I and III), which rapidly inhibit nearby membrane $K^+$ channels to induce depolarization and $Ca^{2+}$ influx[34,42,50]. During more protracted exposures to hypoxia, cationic channels, activated by intracellular $Ca^{2+}$ or cytosolic diffusion of the signaling molecules, may also

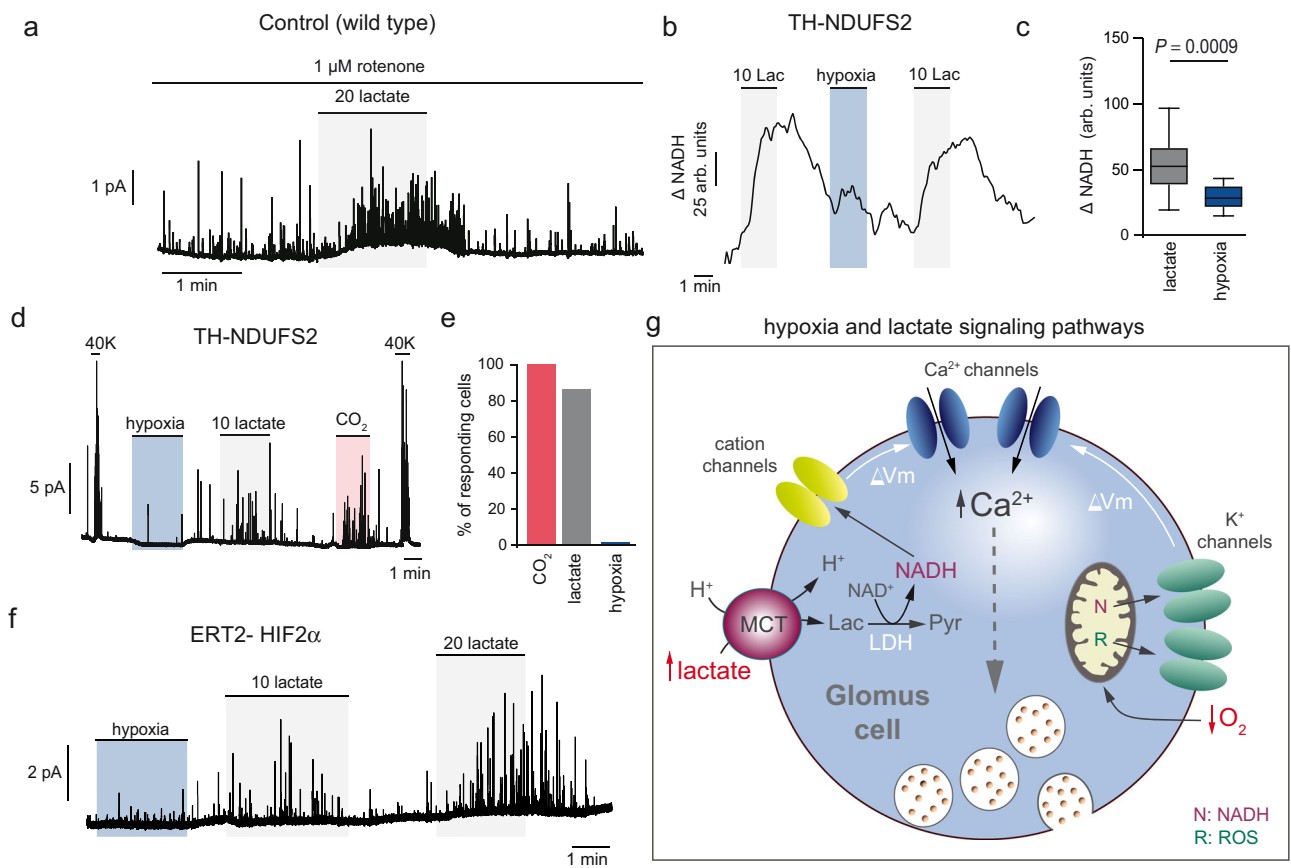

**Fig. 6 Separate lactate- and O₂-sensing mechanisms in glomus cells. a** Amperometric recording illustrating the secretory response to lactate (20 mM, gray) in the presence of rotenone (1 μM). Similar recordings were obtained in 7 cells from 3 mice. **b, c** NADH signals induced by lactate (Lac, 10 mM; gray) in a hypoxia (Hx, O₂ tension ≈15 mm Hg; pale blue)-insensitive glomus cell dispersed from a Ndufs2-deficient CB. Box plots representing the distribution of changes in NADH signals (ΔNADH) in 22 cells from 3 mice exposed to the two stimuli. In these experiments, all cells responded to lactate (10 mM) but only 40% of the cells showed some responsiveness to hypoxia (O₂ tension ≈15 mm Hg). Mean ± SEM values are: 10 mM lactate: 52.5 ± 4.3; hypoxia: 28.2 ± 3.2. *P* value calculated by two tails, unpaired *t* test is indicated. The boxplot represents the median (middle line), 25th, 75th percentile (box), and largest and smallest values extending no further than 1.5 × interquartile range (whiskers). Source data are provided as a Source data file. **d, e** Representative amperometric recordings from an Ndufs2-deficient CB slice showing that cells insensitive to hypoxia (pale blue) are activated by lactate (10 mM, gray), hypercapnia (10% CO₂, pale red) and K⁺ (40 mM)-induced depolarization. Similar experiments (*n* = 6) were performed in CBs from 3 mice. None of the cells from these mice showed a secretory response to hypoxia. **f** Secretory responses to lactate (gray) in a hypoxia-insensitive (pale blue) glomus cell deficient of the *Epas1* (coding Hif2α) gene. Similar experiments (*n* = 2) were performed in CBs from two mice. **g** Schematic representation of the major compartmentalized steps in acute oxygen and lactate sensing by carotid body glomus cells. The decrease of oxygen (hypoxia) is detected by mitochondria, giving rise to the production of NADH (N) and reactive oxygen species (R) which inhibit nearby membrane K⁺ channels to induce depolarization. On the other hand, lactate is transported into the cells by MCTs and rapidly converted to pyruvate with the production of NADH, which activates membrane cation channels to produce cell depolarization (an effect that may be facilitated by intracellular acidification). Pyruvate can also increase the production of reactive oxygen species at the mitochondria and in this manner contribute to the activation of glomus cells. Both hypoxia and lactate increase cytosolic [Ca²⁺] and induce transmitter release. MCT monocarboxylate transporter, LDH lactate dehydrogenase, ΔVm membrane depolarization. Arbitrary units (arb. units).

secondarily contribute to this response[64]. Here we have shown that lactate activation of glomus cells depends on similar signals (NADH and ROS), although NADH is produced in the cytosol as a consequence of imported lactate and its rapid conversion to pyruvate, and ROS are primarily generated at the mitochondrial matrix. To further determine whether hypoxia and lactate share similar sensing/signaling mechanisms, we tested the effect of rotenone, a blocker of MCI that induces glomus cell secretion and occludes any further effect of hypoxia[65]. We observed that rotenone does not prevent the powerful glomus cell activation by lactate (Fig. 6a). Moreover, while responsiveness to hypoxia was practically abolished in mice with ablation of the Ndufs2 gene, which codes a 49 KDa protein essential for MCI assembly and function[42,50], activation of the Ndufs2-deficient and O₂-insensitive glomus cells by lactate (NADH accumulation and catecholamine release) remained unaltered (Fig. 6b–e). Moreover, CB

cells from mice deficient of Hif2α, which have a normal MCI but strong inhibition of responsiveness to acute hypoxia due to down-regulation of atypical MCIV mitochondrial subunits[34], also showed robust responses to lactate (Fig. 6f). These data indicate that lactate and oxygen sensing are mediated by separate mechanisms, although they share compartmentalized signals (rise of cytosolic NADH/NAD⁺ ratio and ROS) that induce depolarization, Ca²⁺ influx, and glomus cell transmitter release (Fig. 6g). In support of this notion, we found that between 65 and 90% of glomus cells that responded to hypoxia also responded to lactate (Fig. 7a–c; see Fig. 2d–f). Moreover, both the NADH signal and the secretory response to hypoxia were strongly potentiated by simultaneous application of extracellular lactate (Fig. 7d–g).

In sum, we show here that the carotid arterial chemoreceptors play a major role in lactate homeostasis, a physiological process of broad functional and medical relevance. CB chemoreceptor cells

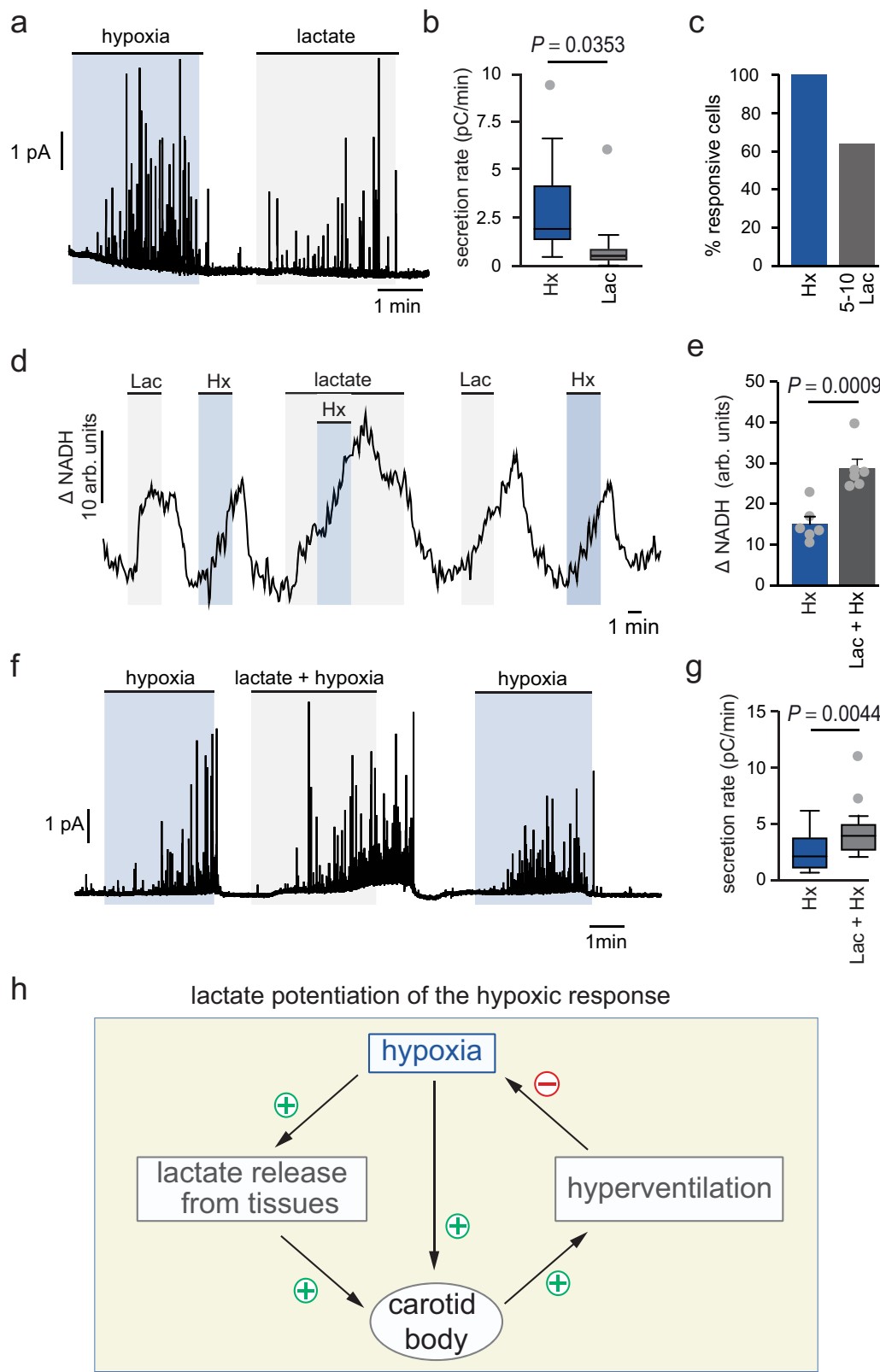

are activated by systemic hypoxia and thereby reduce the intensity of hypoxia-induced lactatemia. In addition, glomus cells are lactate sensors because they efficiently take up and metabolize lactate to increase the cytosolic NADH/NAD$^+$ ratio and mitochondrial ROS production, which modulate the activity of membrane ion channels. Both lactate and hypoxia, although sensed by separate mechanisms, converge on producing cell depolarization and opening of Ca$^{2+}$ channels (see Fig. 6g) and, therefore, have additive effects on glomus cells. The potentiation of CB activation by lactate during hypoxia facilitates a rapid reflex ventilatory compensation of hypoxemia and, secondarily, blunts peripheral lactate release (Fig. 7h). This adaptive response could minimize potential deleterious effects due to lactate release (e.g., metabolic acidosis) or excessive lactate uptake by cells. CB

**Fig. 7 Potentiation of the hypoxic response of glomus cells by lactate. a** Representative amperometric recording of a glomus cell responding to hypoxia (pale blue) and lactate (gray). **b, c** Box plot representing the distribution of the secretion rates in response to each stimuli (**b**) and percentage of cells that responded to hypoxia and lactate (**c**). Mean ± SEM values are: hypoxia: 2839 ± 553 fC/min, $n = 18$; 5–10 mM lactate: 938 ± 546 fC/min, $n = 14$; (13 mice). $P$ values calculated by two tails, unpaired $t$ test are indicated. **d** NADH signal recorded in wild type glomus cells exposed to lactate (10 mM, gray), hypoxia ($O_2$ tension ≈15 mm Hg, pale blue) or both. **e** Bar diagram and scatter plot of changes in NADH ($\Delta$NADH) in response to hypoxia (pale blue) and hypoxia plus lactate (gray) ($n = 6$ experiments from 5 mice). $P$ value calculated by two tails, unpaired $t$ test is indicated. **f, g** Secretory response monitored by amperometry in wild type glomus cells exposed to hypoxia ($O_2$ tension ≈15 mm Hg, pale blue) or lactate (10 mM, gray) plus hypoxia. Values of secretion rates (pC/min) in hypoxia and hypoxia plus lactate are represented by the box plots ($n = 20$ cells from 14 mice). Mean ± SEM values are: Hypoxia (blue) = 2460 ± 512 fC/min; hypoxia + lactate (gray) = 4172 ± 672 fC/min. $P$ value calculated by two tails, paired $t$ test is indicated. The boxplots in (**b, g**) indicate median (middle line), 25th, 75th percentile (box), and largest and smallest values extending no further than 1.5 × interquartile range (whiskers). Data beyond the end of the whiskers (outliers) are plotted individually (gray). Source data are provided as a Source data file. **h** Scheme illustrating potentiation of the CB hypoxic response by lactate and role of CB in lactate homeostasis. + (activation), − (inhibition). Arbitrary units (arb. units).

activation by lactate may also contribute to hyperventilation and lactate homeostasis during physical exercise. The potential side-effects of CB resection or denervation on lactate homeostasis should be considered, as this surgical procedure is used in tumor surgery[66] and is under experimental scrutiny for the treatment of sympathetically mediated diseases[67].

## Methods

**Experimental mice and rat models**. Most of the experiments were performed in wild type adult mice, 2–3-month old without establishing differences in sex. Some experiments were performed in TH-NDUFS2, TH-HIF2α, and ERT2-HIF2α mice, which have previously been used in our laboratory[33,34,42]. In addition, we used the TRPC5 (129S1/SvlmJ-Trpc5[tm1.1Ciph]/J) and TRPC6 (B6J;129S8-$Trpc6^{tm1Lbi}$/ Mmjax) knockout mice purchased from The Jackson Laboratory. Double TRPC5/6 knockout mice were generated in our laboratory. Genotyping of TRPC5, TRPC6, and TRPC5/6 mice was done by PCR using different primers to detect each allele (trpc5 + (WT), 35594 5′-CATCAGTGTTTCTTGCTGCAC-3′ and 35595 5′-GTAG CCCCCTTTCGACTTTC-3′; trpc5- (KO), 35594 and 35596 5′-GCATACTCTT GGGCTCTTTTCA-3′); (trpc6 + (WT), 20556 5′-TCTTTATGCAATCGCTGTGG-3′ and 20557 5′-GCTAGTCT-TCCTGCAATCCA-3′; trpc6- (KO), 20635 5′-TCTA TTAACACTCAACTGGCACCT-3′ and oIMR7415 5′-GCCAGAGGCCACTTG TGTAG-3′) (see Supplementary Table 1 for a complete list of the primers used). Uncropped and unprocessed scans of all the gels done in this study are provided in the Source Data file. To reduce the number of animals used, in vivo experiments (plethysmography and lactate measurements) were performed before animals were sacrificed by intraperitoneal administration of a lethal dose of sodium thiopental (120–150 mg/kg) to dissect tissues for in vitro experiments. A few experiments were performed on 1.5–2 months old Wistar rats. Mice and rats were housed in a controlled environment with a 12-h light-dark cycle, 20–24 °C ambient temperature and without exceeding 55% humidity. All procedures were approved by the Institutional Committee of the University of Seville for Animal Care and Use (2012PI/ LB02 and 22-09-15-332). Handling of the animals was conducted in accordance with the European Community Council directives 86/609/EEC, and 2010/63/EU for the Care and Use of Laboratory Animals.

**Preparation of carotid body slices and dispersed glomus cells**. Mouse and rat CB slices used for amperometric and microfluorimetric recordings were prepared following procedures developed in our laboratory[68,69]. Briefly, carotid bifurcations were extracted, and CBs removed and included in 1% (w/v) low melting point agarose in PBS. 150 μm thick CB sections were cut with a vibratome (VT1000S, Leyca) in cold Tyrode 0 $Ca^{2+}$ solution. Mouse CB slices were treated with an enzymatic solution (PBS pH 7.4 supplemented with 50 μM $CaCl_2$, 0.6 mg/ml collagenase II (Sigma), 0.27 mg/ml trypsin (Sigma) and 1.25 U/ml porcine elastase (Calbiochem) for 5 min at 37 °C. Finally, slices were washed with PBS and cultured in DMEM (0 glucose)/DMEM-F12 (GIBCO) medium (3:1) supplemented with 100 U/ml penicillin (Bio-Witthaker), 10 mg/ml streptomycin (Bio-Witthaker), 2 mM L-glutamine (BioWhittaker), 10% fetal bovine serum (Gibco), 84 U/L insulin (Actrapid ®), and 1.2 U/ml erythropoietin (Binocrit ®) at 37 °C in a 5% $CO_2$ incubator for 24–48 h prior to use. Dispersion and culture of mouse and rat glomus cells used for microfluorimetric recordings and patch clamp experiments, were obtained as described previously by our laboratory[70]. Dissected CBs were incubated for 20 min at 37 °C in the same enzymatic solution described for slices. Then, CBs were stretched with needles and incubated at 37 °C for 5 additional min. Thereafter, cells were dispersed by pipetting, centrifuged, suspended in the same culture medium used for slices (without erythropoietin) and plated on glass cover slips treated with poly-L-lysine (Sigma) and maintained at 37 °C in a 5% $CO_2$ incubator for 24 h.

**Preparation of adrenal medulla slices, dispersed chromaffin cells, and superior ganglion neurons**. Mouse adrenal medulla (AM) slices used for amperometric recordings were prepared following procedures adapted to our

laboratory[69]. Animals were sacrificed and adrenal glands quickly removed and placed on ice-cooled PBS. After removing the capsule, glands were included in low melting point agarose (47 °C) and 200–250 μm thick slices were cut following a protocol similar to that of CB slices. Adrenal medulla slices were then washed with the recording solution (117 NaCl, 4.5 KCl, 23 $NaHCO_3$, 1 $MgCl_2$, 1 $CaCl_2$, 5 glucose and 5 sucrose), bubbled with carbogen (5% $CO_2$ and 95% $O_2$) at 37 °C for 30 min and maintained in cold PBS bubbled with carbogen. Slices were used for experiments during 4–5 h after preparation.

Isolated mouse chromaffin cells were prepared as previously described[56]. Briefly, after removing adrenal glands cortex, medullas were minced with razor blades and resuspended in 3 ml of extraction solution containing 425 U/ml collagenase type IA (Sigma, St. Louis, MO) and 4–5 mg of bovine serum albumin (Sigma) and incubated at 37 °C for 30 min. After this first digestion, cell suspension was centrifuged at room temperature for 3–4 min at 165 × g, the supernatant was removed, and the pellet resuspended in 3 ml of PBS containing 7650 U/ml of trypsin (Sigma) and 425 U/ml of collagenase type IA and incubated at 37 °C for 10 min. To stop the digestion, 10 ml of DMEM culture medium (Invitrogen, Carlsbad, CA) supplemented with 1% (v/v) penicillin (10,000 U/ml)/streptomycin (10,000 μg/ml) (BioWhittaker, Velviers, Belgium), 2 mM L-glutamine (BioWhittaker), and 10% (v/v) fetal bovine serum (Invitrogen) were added and centrifuged at room temperature for 3–4 min at 165 × g. The pellet was resuspended in 100–150 μl of culture medium and plated on glass coverslips treated with poly-L-lysine (1 mg/ml) (Sigma). Cells were then incubated at 37 °C on a 5% $CO_2$ incubator. Isolated cells were used for experiments up to 4–5 h after dispersion.

For dispersion and culture of mouse SCG neurons we followed, with slight variations, a procedure previously described[71]. Briefly, ganglia were cleaned and minced in Leibowitz (L-15) medium and incubated for 15 min in collagenase- and 30 min in trypsin-containing Hanks-balanced saline solution buffered with HEPES. After gentle mechanical disruption, dissociated cells were plated on 35 mm laminin-coated dishes and maintained at 37 °C and 5% $CO_2$ in growth medium (L-15) containing: $NaHCO_3$ 24 mM, FCS 10%, D-glucose 38 mM, L-glutamine 2 mM, penicillin 5000 IU + streptomycin 5000 IU 2.3%, NGF 50 ng/ml). Recordings were made 24–48 h after plating.

**Plethysmography**. To study respiratory function, awake unrestricted mice were placed inside plethysmography chambers (EMKA Technologies) following a procedure adapted to our laboratory[69]. Chambers were perfused with normal air (21% $O_2$, normoxia), 10% $O_2$ (hypoxia), or 5% $CO_2$ (hypercapnia). The hypoxic stimulus was maintained during 5 minutes once $O_2$ percentage reached 10% and the hypercapnic stimulus was maintained during 1 min when $CO_2$ percentage reached 5% $CO_2$. Both $O_2$ and $CO_2$ tensions were continuously monitored and recorded during the experiments.

**Measurement of blood lactate**. Lactate plasma concentration was determined using the Lactate Plus Meter, a hand-held testing device (Nova Biomedical). Two control solutions (1–1.6 and 4–5.4 mM lactate), provided by the supplier, were used to calibrate the device. To determine blood lactate concentration, a small drop of blood obtained from the mice tail after a lancet puncture, was placed on the slot of specific strips for lactate quantification. Lactate measurement was performed under normoxia (21% $O_2$), hypoxia (different oxygen concentrations were used: 15% $O_2$ and 10% $O_2$) and hypercapnia (5% $CO_2$). All the measurements were done in a chamber designed for chronic hypoxia, where gas concentration was easily changed and monitored by specific $O_2$ and $CO_2$ sensors.

**Recording solutions**. For in vitro recordings, either CB, AM, or SCG dispersed cells or CB or AM slices, were transferred to the recording chamber and continuously perfused with a control extracellular solution containing, in mM: 117 NaCl, 4.5 KCl, 23 $NaHCO_3$, 1 $MgCl_2$, 2.5 $CaCl_2$, 5 glucose, and 5 sucrose, at ≈35 °C. In solutions with 40 mM $K^+$ or different lactate/pyruvate concentrations NaCl was replaced equimolarly with KCl, sodium L-lactate or sodium pyruvate respectively. α-ketobutyrate (αKB) was added to the external solution at the indicated

concentrations. When N-methy-D-glucamine (NMDG) was used extracellular $Na^+$ was also equimolarly reduced. The "normoxic" solution was bubbled with a gas mixture of 20% $O_2$, 5% $CO_2$, and 75% $N_2$ ($O_2$ tension ≈145 mm Hg). The "hypoxic" solution was bubbled with 5% $CO_2$, and 95% $N_2$ to reach an $O_2$ tension of ≈15 mm Hg in the recording chamber. The "hypercapnic" solution was bubbled with 20% $CO_2$, 20% $O_2$, and 60% $N_2$. Osmolality of solutions was ≈300 mOsml/kg and the pH 7.4. All the pharmacological drugs used (nifedipine, AR-C155858; 2-APB, Pry3, FTY720, and amiloride)[36,49,55,61–63] were dissolved in stock solutions before they were added directly to the external solution. Concentrations used are indicated in the figures and/or figure legends.

**Amperometric recording of single cell catecholamine secretion in slices.**
Catecholamine secretion from glomus cells or chromaffin cells in CB or AM slices, respectively, was performed following a procedure developed in our laboratory[54,69]. Secretory events elicited by different stimuli were detected with a 10 μm carbon-fiber electrode. Amperometric currents were recorded with an EPC-8 patch clamp amplifier (HEKA Electronics), filtered at 100 Hz and digitized at 250 Hz before storage on computer. Data acquisition and analysis were performed with an ITC-16 interface (Instrutech Corporation) and PULSE/PULSEFIT software (HEKA Electronics). The secretion rate (femtocoulombs (fC)/min) was calculated as the amount of charge transferred to the recording electrode during a given period of time.

**$Ca^{2+}$, NADH, pH, and ROS measurement by single cell microfluorimetry.**
Microfluorimetric measurements in single dispersed glomus cells, chromaffin cells or SCG neurons (intracellular $Ca^{2+}$, pH, and NADH) or cells in CB slices (mitochondrial ROS production) was performed following methods adapted to our laboratory[42,50,70]. To study dispersed glomus cells the system used consists of an inverted microscope (Nikon eclipse Ti) equipped with a 40×/0.60 NA objective, a monochromator (Polychrome V, Till Photonics), and a CCD camera, controlled by Aquacosmos software (Hamamatsu Photonics). To study glomus cells in slices the system used consists of a directed microscope (Olympus, U-TV1x-2) equipped with a 60×/0.90 NA water immersion objective, a monochromator (Polychrome V, Till Photonics), and a CCD camera, controlled by Live Adquisition software (TILL Photonics). For the experiments, dispersed cells or carotid body slices were transferred to the recording chamber and perfused with the solutions described above. All experiments were performed at 35–37 °C

Cytosolic $Ca^{2+}$ was measured in dispersed CB glomus cells loaded at 37 °C for 30 min with 4 mM Fura2-AM (TefLabs) in serum-free DMEM/F-12. To remove excess Fura2-AM, the coverslip was incubated for 15 min in complete medium and then transferred to the recording chamber with a continuous solution flow (see "Recording solutions"). Alternating excitation wavelengths of 340 and 380 nm were used, and background fluorescence was subtracted before obtaining the F340/F380 ratio[35]. A dichroic FF409-Di03 (Semrock) and a band pass filter FF01-510/84 (Semrock) were used. Cytosolic [$Ca^{2+}$] signals were digitized at a sampling interval of 500 ms.

NADH microfluorimetric measurements (NAD(P)H autofluorescence) were performed in CB, AM or SCG dispersed cells using a non-ratiometric protocol widely used to estimate changes in NAD(P)H levels in CB cells[34,42,47,48]. NADH was excited at 360 nm and measured at 460 nm. The acquisition protocol was designed with a spatial resolution of 4 × 4 pixels, an excitation time of 150 ms, and an acquisition interval of 5 s. A dichroic FF409-Di03 (Semrock) and a band pass filter FF01-510/84 (Semrock) were used. Background fluorescence was subtracted in all the experiments (Arias-Mayenco et al., 2018). In parallel with these experiments, we attempted to measure NADH in dispersed glomus cells using the selective ratiometric sensor Peredox[72]. However, we did not observe an efficient transfection of Peredox-encoding plasmids during the time-period that dispersed glomus cells remained in healthy conditions (~20–30 h).

Changes of intracellular pH were monitored in dispersed glomus cells loaded with 2 μM BCECF-AM (acetoxymethyl ester of 2′,7′-bis (2-carboxyethyl)−5, 6-carboxyfluorescein; Molecular Probes) in PBS at room temperature for 15 min. Alternating excitation wavelengths of 490 and 440 nm were used and recorded at 535 nm. Background fluorescence was subtracted before obtaining the 490/450 ratio[56]. A dichroic Di02-R488 (Semrock) and a band pass filter FF01-520/35 (Semrock) were used.

Rapid changes in ROS production were monitored in glomus cells in CB slices, using redox-sensitive green fluorescent protein (roGFP) probes targeted to either the mitochondrial intermembrane space (IMS) or mitochondrial matrix[73]. To infect CB slices with the adenoviral vector (ViraQuest Inc), freshly prepared slices were incubated for 48 h in complete culture medium supplemented with 2 ml of the different adenoviral roGFP construction to target the specific subcellular compartments: IMS-roGFP (VQAd CMV GDP-roGFP) or matrix-roGFP (VQAd CMV mito-roGFP). For the experiments the infected CB slice, expressing roGFP, was transferred to the recording chamber and bathed with continuous flow of solution (see "Recording solutions"). A dichroic Di02-R488 (Semrock) and a band pass filter FF01-520/35 (Semrock) were used for the experiments. RoGFP was excited at 400 and 484 nm, and emission recorded at 535 nm, allowing ratiometric measurements of rapid and reversible changes in mitochondrial redox state[42,50]. To ensure that the probe worked correctly, a brief pulse of 0.1 mM $H_2O_2$ was applied at the end of all the experiments to test for rapid maximal oxidation.

**Patch clamp recordings in dispersed carotid body glomus cells and SCG neurons.** Membrane potential was recorded from dispersed mouse glomus cells and SCG neurons using the perforated patch configurations of the patch clamp technique as adapted in our laboratory[68,70]. Patch clamp pipettes (2–4 MΩ) were pulled from capillary glass tubes with a horizontal pipette puller P-1000 (Sutter instruments) and fire-polished with a microforge MF-830 (Narishige). Current-clamp recordings were obtained with an EPC-7 amplifier (HEKA Electronik). The signal was filtered (3–10 kHz), subsequently digitized with an analog/digital converter ITC-16 (Instrutech Corporation) and finally sent to the computer. Data acquisition and storage were performed using the Pulse/Pulsefit software (HEKA Electroniks) at a sampling interval of 4 ms. Data analysis were performed using the Igor Pro Carbon (Wavemetrics) and Pulse/Pulsefit (HEKA Electronik) programs. For perforated patch experiments the pipette solution contained (in mM): 70 $K_2SO_4$, 30 KCl, 2 $MgCl_2$, 1 EGTA, 10 HEPES, pH 7.2. Amphotericin B (240 μg/ml) was added to this solution. Bath solutions composition was described in "Recording solutions".

**Immunohistochemical analysis.** For Immunofluorescent studies, mice were perfused first with PBS and then with 4% paraformaldehyde in PBS before tissue dissection. Carotid bifurcations and adrenal glands were fixed with 4% paraformaldehyde in PBS for 2 h, cryoprotected overnight with 30% sucrose in PBS, and embedded in OCT (Tissue-Tek). Tissue sections of 8 μm were obtained with a cryostat (Leyca). These sections were incubated first with primary antibodies overnight at 4 °C, then with fluorescent secondary antibodies at room temperature for 2 h. The primary antibodies used were: MCT1 (rabbit, 1:200 dilution, NBP1-59656, Novus Biologicals), MCT2 (rabbit, 1:500 dilution, PA5-77498, Thermo-Fisher Scientific), MCT4 (rabbit, 1:100 dilution, 22787-1-AP, Proteintech), TH (sheep, 1:200 dilution, AB1542, Millipore/Merck), and GFAP (chicken, 1:500, ab4674, Abcam). The secondary antibodies were: Alexa Fluor 568 (anti-rabbit, 1:400 dilution, A-11011, ThermoFisher Scientific), Alexa Fluor 488 (anti-sheep, 1:400 dilution, A-11015, ThermoFisher Scientific), and Alexa Fluor 488 (anti-chicken, 1:400 dilution 103-545-155 Jackson ImmunoResearch). In addition, nuclei were labeled with 4′,6′-diamidino-2-phenylindole (DAPI, 1:1000 dilution). Immunofluorescent images were obtained using Nikon A1R+ confocal microscopy. When necessary, orthogonal projection from Z-stack images was also presented in order to study potential colocalization between two proteins in the same cell.

**Statistical analysis.** Normality of the data sets obtained in the experiments was tested with the Shapiro–Wilk test. When necessary, a log transformation was performed to normalize the distribution prior to parametric analyses. Except in a few cases, specified in the figure legends, data had a normal distribution and were described as mean ± SEM with the number ($n$) of experiments indicated. Data from two groups were analyzed with either a two tails $t$ test or a paired two tails $t$ test. Data with multiple groups were analyzed by ANOVA or RM-ANOVA followed by a post-hoc Tukey's test. For graphical representation of the data we used bar diagrams, with indication of the mean ± SEM, and a scatter plot of the data points superimposed. For the sake of clarity, we used for graphical representation box plots, with indication of median, quartiles, and outliers, in panels where all data sets had $n ≥ 10$ data points. In these last cases mean ± SEM values are given in the figure legends. For data without a normal distribution the analysis of statistically significant differences between distributions was done with non-parametric tests (Mann–Whitney rank-sum test or Kruskal–Wallis test, followed by a post-hoc Tukey's test). For graphical representation of non-parametric data, we used box plots with indication of median, quartiles, and outliers. Mean ± SEM values in these data sets are also given in the figure legends. Statistical analyses were done using Prism Version 8.2.1. (279) for MacOS. A $P < 0.05$ or smaller was considered statistically significant.

**Reporting summary.** Further information on research design is available in the Nature Research Reporting Summary linked to this article.

## Data availability
Data generated or analyzed over the course of this study are included within the paper. The data that support the findings of this study are available from the corresponding author upon reasonable request. Source data are provided with this paper.

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

## Acknowledgements

This research was supported by the Spanish Ministries of Science and Innovation and Health (SAF2016-74990-R), and the European Research Council (ERC Advanced Grant PRJ201502629). H.T.-T. received a predoctoral fellowship (FPI program) from the Spanish Government. We thank Drs. Vicky Bonilla-Henao, Ana Muñoz-Cabello, and Olalla Colinas, as well as Mrs./Mr. Paula Garcia-Flores, Helia Sarmiento, Blanca Jiménez-Gómez, and Mr. Antonio Bejarano for help with some of the experiments. We also thank IBiS staff for technical assistance.

## Author contributions

H.T.-T., P.O.-S., and L.G. performed the experiments and analyzed the data. P.O.-S., H.T.-T., and J.L.-B. designed the study and contributed to generating the draft of the manuscript. J.L.-B. coordinated the project and the writing of the paper.

## Competing interests

The authors declare no competing interests.
