## [Peer Review File · Nature Communications]

Reviewer comments, first round

Reviewer #1 (Remarks to the Author):

This is an interesting study confirming previous findings by this groups and other investigators that glomus cells of the carotid body may sense lactate. The novel component of the study that the authors provided novel evidence that lactate sensing in the carotid body induces ventilatory response to lactate blunting peripheral lactate release and that lactate sensing is independent of hypoxia sensing. Several major findings are reported here: 1) This group has recently shown that HIF-2a plays a major role in hypoxia sensing by glomus cells. Here they showed that HIF-2a in glomus cells plays a major role in lactate and hypoxia sensing, but not CO₂ sensing; 2) MCT2 plays a role in lactate transport in glomus cells and pyruvate. Lactate increases cytosolic Ca²⁺, similar to hypoxia and showed that it is metabolized to pyruvate by LDH with production of NADH; 3) Lactate increased mitochondrial matrix ROS production (in contrast to hypoxia). Furthermore, experiments with rotenone, a blocker of MCI and with Ndufs2 deficiency (a critical protein for MCI), which block effects of hypoxia, showed no difference in response to lactate. This experiment showed a fundamental difference in mechanism of lactate response vs hypoxia response; 4) Electrophysiological effects of lactate were abolished by 2-APB, a non-selective TRP channel blocker. They have also tested CB glomus cells from TRPC5 KO mice and showed no effect of KO on lactate responses .

Major critique

The paper showed that glomus cells sense lactate by a different mechanism than they sense hypoxia. However, the mechanism(s) of lactate sensing and downstream pathways are not clearly identified. Is lactate sensed by the same population of glomus cells which sense hypoxia? Is the MCT2 critical and only transporter? It looks like mitochondrial ROS production is a critical mechanism, but how it works if not through MCI? The investigators implicated TRP channels but they used only one non-selective blocker. They also tested somewhat randomly selected TRPC5 with negative results. Finally, how does HIF-2a fits with the rest of the story?

In other words, the authors' findings that glomus cells sense lactate, independent of hypoxia and hypercapnia, and that this sensing may play an important role in ventilatory compensation for lactic acidosis, is very important. However, the attempt to outline mechanisms of lactate sensing is somewhat fragmentary.

Reviewer #2 (Remarks to the Author):

The manuscript by Torrrs-Torrelo et al. deals with the mechanisms of lactate sensing by the carotid body (CB), a mammalian sensory organ that is now recognized as a polymodal metabolic sensor of several blood chemicals, and is involved in the control of breathing and autonomic functions. Though lactate sensing by the CB has been recently reported by a couple of other research groups, the mechanism of action was largely unknown and the initial proposition that it occurred via an olfactory receptor (Olf78), uniquely expressed in CB chemoreceptor (glomus) cells, has been challenged by studies in this and another laboratory. The novelty of the present study is that it uncovers for the first time mechanisms of lactate sensing by CB glomus cells, including key steps in the transduction pathway that are independent of Olf78; the study also provides additional data on the role of the CB in lactate homeostasis. In general, the studies were well executed and performed by a leading and world-renowned laboratory that has contributed significantly to our understanding of the cellular mechanisms of CB chemotransduction. The present study is a logical extension of their previous elegant work, and also an important one that utilizes an impressive array of experimental techniques including plethysmography, electrophysiology on isolated CB cells and tissue slices, carbon fiber amperometry, and fluorescence imaging tools applied to both wild-type and transgenic mouse models. The final

product is yet another significant contribution from this group to the CB field and adds significantly to our understanding of the general mechanisms by which the CB contributes to respiratory and circulatory homeostasis in health and disease. Nonetheless, I believe the general significance and impact of the study can be considerably improved if the authors addressed the comments indicated below.

Critique

1. An issue raised by the study is whether or not CB glomus cells can be considered as 'specialized' lactate sensors analogous to their established roles as specialized PO₂ and PCO₂/H⁺ sensors. If so, it would be of great interest, expanding the notion of CB chemoreceptors as specialized, polymodal metabolic sensors involved in respiratory and cardiovascular homeostasis. While the authors have provided strong evidence that CB glomus cells can act as lactate sensors (within the physiological range of blood lactate levels seen in normal and pathophysiological conditions), the question regarding specificity or selectivity of this response was not adequately addressed. While they provide immunohistochemical and functional evidence for monocarboxylate transporters (MCT) as well as LDH involvement in lactate sensing by glomus cells, these are ubiquitous markers found in many different cell types. It is not clear in Fig. 2b whether other CB cell types stain for MCT and if so, do they respond to lactate? Perhaps, even more interesting, is whether or not other sympathoadrenal TH⁺ positive neural-like cells from a similar embryonic lineage as glomus cells are also lactate sensors. Sympathetic neurons would be an excellent control in this regard given that, unlike glomus cells they lack O₂ and CO₂ chemosensitivity, but similar to glomus cells they express TH, release catecholamines when depolarized, and are involved in the regulation of autonomic functions. Since these neurons likely express both MCT and LDH, the question arises whether lactate similarly increases NADH autofluorescence and mitochondrial ROS production, leading to membrane depolarization in these cells? A negative answer would lend support to the idea of glomus cells as 'specialized' lactate sensors.

2. Using a variety of experimental models, the authors clearly show the mechanisms of O₂ and lactate sensing by CB glomus cells are distinct, though there is some overlap in the signals involved in the transduction schema. For example, both stimuli lead to an increase in NADH and ROS generation at the mitochondrial intermembrane space (IMS), though ROS levels appear to move in opposite directions in the mitochondrial matrix. For lactate sensing they propose the increase in NADH or NADH/NAD⁺ ratio and ROS lead to membrane depolarization and Ca²⁺-dependent secretion due to activation of a standing (Na⁺ permeant) cation conductance. Yet, a similar increase in these signals is proposed to mediate hypoxia sensing in the same cells but via inhibition of background K⁺-selective TASK1/3 ion channels. How do the authors explain the convergence of these identical signals onto two different channel types in the same cell and particularly, how is selectivity towards their respective channel type achieved when either lactate or hypoxia is present alone?

3. The potential role of TRPC5 channels in mediating the lactate-induced membrane depolarization and glomus cell secretion was ruled out by the authors in experiments using TRPC5- deficient glomus cells obtained from knockout mice. However, TRPC3 channel expression is higher than TRPC5 in CB glomus cells (text refs. 30,44) and TRPC3 has been implicated in sensing of other glycolytic substrates (e.g. glucose) in hypothalamic neurons (Diabetes 2017 Feb; 66(2): 314-324). It would be interesting to know if putative 'selective' TRPC3 blockers (e.g. Pyr 3/10; see Cells 2018 July 7(7):83) inhibit lactate sensing in CB glomus cells.

Minor comments:

1. In Abstract line 8, the authors claim "...the carotid body is essential for lactate homeostasis". Presumably this is based largely on the observation of elevated hypoxia-induced lactatemia in CB-deficient mice, i.e. TH-HIF2a and ERT2-Cre-HIF2a mice, where the CB fails to develop and/or generates a defective HVR. However, other extra-CB cells/tissues in the peripheral and central nervous systems are affected in these HIF2a-deficient mice. Can the authors comment on whether CB denervation prevents the elevated hypoxia-induced lactatemia?
2. In Fig. 3 there seems to be broad variability in the latency of the glomus cell depolarizing response to lactate. How does this compare to the latency of the hypoxia response?
3. In Extended Data Fig. 3b, should the second bin read (Lac + a-KB)?

Reviewer #3 (Remarks to the Author):

A new homeostatic function of the carotid body (CB) is proposed, which is the sensing of blood lactate leading to a upregulation of ventilation. After showing inordinate blood lactate levels in response to hypoxia in two CB-deficient mice transgenic lines, the study focuses on glomus cells in vitro. The main findings are:

- lactate induces the release of catecholamines, a phenomenon sensitive to Cd²⁺
- MCT2 is expressed
- lactate increases cytosolic Ca²⁺ and acidifies
- lactate increases UV autofluorescence, sensitive to AR-C1558585 and reverted by pyruvate
- lactate and pyruvate increase mitochondrial ROS.
- lactate depolarizes cells, which requires extracellular Na⁺
- In hyperpolarized cells, lactate triggers action potentials, reverted by pyruvate and inhibited by 2-APB
- The increase in autofluorescence and secretory response induced by lactate is insensitive to pharmacological and genetic disabling of mitochondria
- The secretory response to hypoxia is stronger in the presence of lactate.

From these results, a chain of events is proposed: lactate transport, LDH-mediated conversion into pyruvate leading to elevated NADH/NAD⁺, membrane depolarization, cation channel mediated-depolarization, activation of voltage-sensitive Ca²⁺ channels, and secretion.

This is a potentially important story. I am not aware of any homeostatic system in which blood lactate is the regulated variable, for lactate accumulation is usually regarded as an epiphenomenon. The secretory response of glomus cells to lactate is convincing, but the characterization of the signaling mechanisms seems preliminary. The manuscript is concisely written and well organized. However, I feel that some of the data are not strong enough to support the conclusions as they are, that more experiments are needed to clarify each node of the purported signaling chain, and that some assumptions are not well supported by the literature. There are also technical issues to be addressed. Authors are invited to consider the following specific points:

1. Is higher lactate in CB-disabled mice explained by weaker HVR? Otherwise, it may be attributed to general effects of HIF2 α deletion (Macias et al., 2018). Please provide the breathing response to hypoxia of the KO animals as compared to that of controls.
2. Cellular UV autofluorescence is hardly a readout of cytosolic NADH. Autofluorescence does not distinguish between NADH and NADPH, and for that reason is referred to as NAD(P)H autofluorescence, and it does not distinguish either between free and protein-bound nucleotides (which are 99% of the total). Critically, most cellular UV autofluorescence comes from the mitochondrial NADH pool and the rest is cytosolic NADPH, with an almost undetectable contribution of the very small cytosolic NADH pool (e.g. discussed by Shuttleworth et al., *Neurochem. Int.* 2010;). Thus, it is not possible to ascribe whole cell autofluorescence measurements to cytosolic NADH, and the current results do not demonstrate LDH-mediated oxidation of lactate. The increase may well have resulted from mitochondrial NADH accumulation induced by substrate, and/or by Ca²⁺ or acidification. Nowadays cytosolic NADH is routinely determined with precision and unambiguously using genetically-encoded sensors like Peredox, Hex, SONAR, etc.
3. Even if a cytosolic NADH were verified with a genetically-encoded sensor, demonstration of a causal connection to cell depolarization, Ca²⁺ rise and catecholamine secretion needs more experiments. For example independent manipulation of cytosolic NADH by addition of extracellular NADH. Against a major role for cytosolic NADH, pyruvate, which reduces cytosolic NADH, also induced a robust secretory response in glomus cells (Extended Data Fig. 4B).
4. Similar effects of lactate and pyruvate suggest that the secretory response may relate to

mitochondrial fuelling. Even in the presence of glucose, cytosolic pyruvate is often near zero if there is no pyruvate in the medium (e.g. San Martin et al., PLoS One 2014; Contreras-Baeza et al., JBC 2019). This "pre-starvation" hypothesis may explain why both lactate and pyruvate evoked mitochondrial ROS responses and why alpha-ketobutyrate prevented the effect of lactate. A confounding effect of pyruvate shortage could be tested by trying lactate-stimulation in physiological pyruvate (100 μ M). Another effect common to lactate and pyruvate is cytosolic acidification, which is bound to affect matrix pH and mitochondrial metabolism (Azarias et al. J. Neurosci. 2011).

5. The Ca^{2+} response warrants a deeper characterization. Sensitivity to Cd^{2+} is not enough to decreet extracellular Ca^{2+} involvement, as Cd^{2+} inhibits all kinds of channels and transporters. Could acidification contribute to the Ca^{2+} rise, perhaps by engagement of NHE and NCX? That might help to explain the Na^{+} sensitivity of the effect and why pyruvate also induced a secretory response. Is there a contribution of Ca^{2+} stores?

6. There is variability in the onset of the lactate effects, which makes it hard to visualize a causal chain. According to Extended Data Fig. 2, Ca^{2+} rises within 1 minute of lactate exposure but pH starts to drop afterwards. This delay does not support the suggested chain of events. If this variability is not due to perfusion issues, please provide simultaneous measurements of the critical variables.

7. MCTs. Immunocytochemistry of MCTs is notorious for false positive and negatives as commercial antibodies are suspect and epitopes are often masked. It is not clear what is the point of showing MCT2 expression, without reference to MCT1, which is also blocked by AR-C155858. Moreover, most of the staining seems to be cytoplasmic (Mitochondria perhaps?). More informative would be MCT 1/2 immunoblots of plasma membranes.

8. Introduction. The notion that lactate is the final product of glycolysis is correct for cells that export lactate, like white muscle fibers and brain astrocytes, but not for those that consume lactate, like red muscle fibers and neurons (discussed by Barros et al., Neurochem. Res. 2020). Authors should also be aware that the intracellular lactate shuttle is far from "well established" a concept, rather the opposite (Glancy et al., J. Physiol. 2020).

Minor

9. Citations. Reference number 3 is not cited, whereas number 5 relates to tumors, not to healthy tissues. Reference number 6 is a comment on number 3. Please check the paper throughout.

10. Pg. 4, line 8. "did evoked an ventilatory".

11. Pg. 5, line 5. "directly". The pH technique is indirect, direct monitoring is done with lactate or pyruvate sensors.

12. Pg 7, line 7. "signaling. molecules"

L. Felipe Barros

Ms. NCOMMS-20-29211A(R1)
Authors: Torres-Torrelo et al.,

RESPONSE TO THE REVIEWERS' COMMENTS

Authors: We very much appreciate the reviewers for the detailed and insightful evaluation of our manuscript. We have done several new sets of experiments to address all their comments and suggestions. The revised manuscript, which was before in a short report format, is now longer and divided in sections and subheadings, with 7 main figures and 8 supplemental figures. The changes made in the text are marked in red. We have added a new author (Dr. Lin Gao) who has greatly contributed to the new experiments, figures and text. Please note that the time required to finish the revised manuscript has been longer than expected due, among other reasons, to the restrictions in the use of laboratories and delays in the delivery and breeding of experimental animals as a consequence of the Covid-19 pandemics. We hope that the editor and reviewers are satisfied with the revisions made.

Reviewer #1 (Remarks to the Author):

This is an interesting study confirming previous findings by this groups and other investigators that glomus cells of the carotid body may sense lactate. The novel component of the study that the authors provided novel evidence that lactate sensing in the carotid body induces ventilatory response to lactate blunting peripheral lactate release and that lactate sensing is independent of hypoxia sensing. Several major findings are reported here: 1) This group has recently shown that HIF-2a plays a major role in hypoxia sensing by glomus cells. Here they showed that HIF-2a in glomus cells plays a major role in lactate and hypoxia sensing, but not CO₂ sensing; 2) MCT2 plays a role in lactate transport in glomus cells and pyruvate. Lactate increases cytosolic Ca²⁺, similar to hypoxia and showed that it is metabolized to pyruvate by LDH with production of NADH; 3) Lactate increased mitochondrial matrix ROS production (in contrast to hypoxia). Furthermore, experiments with rotenone, a blocker of MCI and with Ndufs2 deficiency (a critical protein for MCI), which block effects of hypoxia, showed no difference in response to lactate. This experiment showed a fundamental difference in mechanism of lactate response vs hypoxia response; 4) Electrophysiological effects of lactate were abolished by 2-APB, a non-selective TRP channel blocker. They have also tested CB glomus cells from TRPC5 KO mice and showed no effect of KO on lactate responses.

Authors: We thank very much the reviewer for his/her detailed evaluation of our manuscript, the positive comments and specific suggestions for improvement. We have taken into consideration all the reviewer's comments and criticisms to prepare the revised version of the manuscript.

Major critique

The paper showed that glomus cells sense lactate by a different mechanism than they sense hypoxia. However, the mechanism(s) of lactate sensing and downstream pathways are not clearly identified.

Authors: We think that the mechanisms underlying lactate sensing by CB glomus cells and the differences with hypoxia sensing are now better explained (pages 10,11). We have added a summary scheme to Fig. 6 (panel g) which is explained in detail in the figure legend.

We have also done new experiments on Hif2a-deficient mice to further illustrate the differences between oxygen and lactate sensing (Fig. 6f). The main steps in lactate sensing by glomus cells are: lactate transport and production of NADH and ROS, activation of a 2-APB-sensitive cationic conductance, membrane depolarization and Ca²⁺-dependent transmitter release. We have added a short comment suggesting that in normal conditions intracellular acidification may also contribute to lactate activation of glomus cells (page 8, lines 3-5). We have performed new experiments using genetically modified mice (TRPC6 KO and TRPC5/6 double KO, Supplementary Fig. 7) and pharmacological tools (Supplementary Fig. 8) to search for the cationic channels mediating lactate activation of glomus cells. However, we have not been able to identify a channel selectively modulated by lactate. We believe that NADH and ROS are promiscuous signals that may redundantly modulate several subtypes of ion channels in glomus cells. These cells are known to express background Na⁺-permeant currents (Carpenter and Peers, 2001; García-Fernández et al., 2007) and several TRP channel classes (Buniel et al., 2003; Zhou et al., 2016; Gao et al., 2017).

Is lactate sensed by the same population of glomus cells which sense hypoxia?

Authors: We have included new data showing that most of the cells that responded to hypoxia (~80% on average, monitored by either an increase in cytosolic Ca²⁺ or by an increase in catecholamine release) also responded to lactate (Figs. 2d-f; Fig. 7a-c) (page 5, lines 13-15; page 11, lines 10-12).

Is the MCT2 critical and only transporter?

Authors: We have performed a more detailed study of the expression of the most relevant MCTs (MCT1, 2 and 4, Halestrap, 2013) in carotid body glomus (O₂-sensitive) and type II cells. Glomus cells do not express MCT1 (at least in sufficient level to be detected by immunocytochemistry) but they express MCT2 and MCT4 (Fig. 3, and Supplementary Figs. 2 and 4). Given the Km value of MCT2 for lactate (~1 mM) it is likely that this MCT is the one mediating lactate uptake in normal physiological conditions. Indeed, AR-C155858 (a non-selective blocker of MCT2 and MCT1) strongly inhibited lactate-induced rise in NAD(P)H autofluorescence (Fig. 4d). However, it is possible that MCT4 (Km>20 mM) contributes to lactate uptake by glomus cells when plasma concentration is very high. As we have noticed before that glomus cells can survive genetic dysfunction of mitochondrial electron transport (Fernandez Agüera et al., 2015) we have also discussed the possibility that MCT4 contributes to lactate release by these MCI-deficient cells, which probably have an accelerated glycolytic flux and intracellular accumulation of lactate (page 6, lines 15-20). Interestingly, it seems that type II cells (of glia-like origin) do not express MCT2 (Fig. 3c; Supplementary Fig. 2) or MCT1 (Supplementary Fig. 4a,c). However, they might express some low levels of MCT4 (Supplementary Fig. 4b, d) (page 6, lines 20-24).

It looks like mitochondrial ROS production is a critical mechanism, but how it works if not through MCI?

Authors: We think that glomus cells have a very active mitochondria with a high basal production of ROS, which rises in the presence of higher levels of substrates (lactate or pyruvate). This is a “constitutive” ROS production probably generated as a consequence of the activity of Krebs cycle dehydrogenases as well as of complexes of the electron transport chain (ETC). Therefore, this “substrate-dependent” increase in ROS production is not only due to MCI activity. Indeed, we have shown that ROS production in glomus cell mitochondrial matrix decreases in response to hypoxia (because the concentration of O₂, one of the substrates required to produce ROS, decreases) and this response is maintained in MCI-deficient mice (Arias-Mayenco et al., 2018). ROS produced in the mitochondrial matrix are

rapidly converted to H₂O₂ (due to the high superoxide dismutase activity) which easily moves out to the intermembrane space and cytosol.

The investigators implicated TRP channels but they used only one non-selective blocker.

Authors: We initially focused on TRP channels because in a previous single-cell sequencing study performed on mouse glomus cells (Zhou et al., 2016), it was found that they have abundant mRNA coding for TRPC5, TRPC3 and TRPM7, as well as beta subunit of the epithelial Na⁺ channels. The high expression of TRPC5 in glomus cells was also confirmed in an independent microarray study performed in our laboratory (Gao et al., 2017). We have now expanded the pharmacological studies on TRP channels. We have tested the effect of Pyr 3 (a blocker of TRPC3 suggested by reviewer 2; Tiapko and Groschner, 2018) (Supplementary Fig. 8a,b), FTY720 (a blocker of TRPM7; Chubanov et al., 2014) (Supplementary Fig.8c), amiloride (a blocker of epithelial Na⁺ channels; Canesa et al., 1996,) (Supplementary Fig. 8d) and the three drugs together (Supplementary Fig. 8 e,f). None of these drugs seemed to clearly inhibit glomus cell activation by lactate. Moreover, in addition to TRPC5, we have now shown that genetic ablation of either TRPC6 or TRPC5 plus TRPC6 have no effect on lactate activation of glomus cells (Supplementary Fig. 7). In addition to the 2-APB experiment (done with amperometry and patch clamp techniques), we show that the lactate-induced glomus cell depolarization and transmitter release are selective (they are not seen in embryologically related chromaffin cells or superior cervical ganglion neurons; Supplementary Fig. 6a-f) and abolished by removal of extracellular Na⁺ (Fig. 5). Therefore, our conclusion is that there is not a “lactate-sensing” channel but that lactate-induced NADH and ROS may redundantly modulate several subtypes of cationic channels in glomus cells (page 10, lines 1-10).

They also tested somewhat randomly selected TRPC5 with negative results.

Authors: As indicated above, TRPC5 was initially selected because in two independent studies (Zhou et al., 2016; Gao et al., 2017) it was reported to be highly expressed in mouse CB cells (page 9, lines 5-8 from the bottom). The TRPC5 data have been completed with the study of mice lacking TRPC6, a channel type expressed in CB cells (Buniel et al., 2003) that has been reported to be involved in oxygen sensing in pulmonary myocytes (Weismann et al., 2006) (page 9, last three lines and page 10, lines 1-3)

Finally, how does HIF-2a fits with the rest of the story?

Authors: Hif2a by itself does not seem to have a primary role in lactate sensing. However, Hif2a-dependent expression of atypical MCIV subunit isoforms is required for acute O₂-sensing and glomus cells from Hif2a-deficient mice have a marked selective inhibition of responsiveness to hypoxia (Moreno-Domínguez et al., 2020). This is the reason why Hif2a-deficient mice show a strong increase in plasma lactate during hypoxia, as their CB is not able to trigger a compensatory ventilatory response (Fig. 1d). To clarify this point we have performed new experiments to show that Hif2a-deficient CB cells, are insensitive to hypoxia but can be activated by lactate (Fig. 6f) (page 11, lines 5-7).

In other words, the authors' findings that glomus cells sense lactate, independent of hypoxia and hypercapnia, and that this sensing may play an important role in ventilatory compensation for lactic acidosis, is very important. However, the attempt to outline mechanisms of lactate sensing is somewhat fragmentary.

Authors: We thank again the reviewer for his/her helpful comments and hope that he/she is satisfied with our new experimental data and more detailed explanation of the lactate

signaling pathway, as well as of the differences between acute oxygen and lactate sensing.

Reviewer #2 (Remarks to the Author):

The manuscript by Torrrs-Torrelo et al. deals with the mechanisms of lactate sensing by the carotid body (CB), a mammalian sensory organ that is now recognized as a polymodal metabolic sensor of several blood chemicals, and is involved in the control of breathing and autonomic functions. Though lactate sensing by the CB has been recently reported by a couple of other research groups, the mechanism of action was largely unknown and the initial proposition that it occurred via an olfactory receptor (Olf78), uniquely expressed in CB chemoreceptor (glomus) cells, has been challenged by studies in this and another laboratory. The novelty of the present study is that it uncovers for the first time mechanisms of lactate sensing by CB glomus cells, including key steps in the transduction pathway that are independent of Olf78; the study also provides additional data on the role of the CB in lactate homeostasis. In general, the studies were well executed and performed by a leading and world-renowned laboratory that has contributed significantly to our understanding of the cellular mechanisms of CB chemotransduction. The present study is a logical extension of their previous elegant work, and also an important one that utilizes an impressive array of experimental techniques including plethysmography, electrophysiology on isolated CB cells and tissue slices, carbon fiber amperometry, and fluorescence imaging tools applied to both wild-type and transgenic mouse models. The final product is yet another significant contribution from this group to the CB field and adds significantly to our understanding of the general mechanisms by which the CB contributes to respiratory and circulatory homeostasis in health and disease. Nonetheless, I believe the general significance and impact of the study can be considerably improved if the authors addressed the comments indicated below.

Authors: We thank very much the reviewer for his/her detailed evaluation of our manuscript, the positive comments and specific suggestions for improvement. We have performed new experiments and taken into consideration all the reviewer's comments and criticisms to prepare the revised version of the manuscript.

Critique

1. An issue raised by the study is whether or not CB glomus cells can be considered as 'specialized' lactate sensors analogous to their established roles as specialized PO₂ and PCO₂/H⁺ sensors. If so, it would be of great interest, expanding the notion of CB chemoreceptors as specialized, polymodal metabolic sensors involved in respiratory and cardiovascular homeostasis. While the authors have provided strong evidence that CB glomus cells can act as lactate sensors (within the physiological range of blood lactate levels seen in normal and pathophysiological conditions), the question regarding specificity or selectivity of this response was not adequately addressed. While they provide immunohistochemical and functional evidence for monocarboxylate transporters (MCT) as well as LDH involvement in lactate sensing by glomus cells, these are ubiquitous markers found in many different cell types. It is not clear in Fig. 2b whether other CB cell types stain for MCT and if so, do they respond to lactate?

Perhaps, even more interesting, is whether or not other sympathoadrenal TH⁺ positive neural-like cells from a similar embryonic lineage as glomus cells are also lactate sensors. Sympathetic neurons would be an excellent control in this regard given that, unlike glomus

cells they lack O₂ and CO₂ chemosensitivity, but similar to glomus cells they express TH, release catecholamines when depolarized, and are involved in the regulation of autonomic functions. Since these neurons likely express both MCT and LDH, the question arises whether lactate similarly increases NADH autofluorescence and mitochondrial ROS production, leading to membrane depolarization in these cells? A negative answer would lend support to the idea of glomus cells as 'specialized' lactate sensors.

Authors: Following the reviewer suggestions we have performed experiments on superior cervical ganglion (SCG) neurons and adrenal medulla (AM) chromaffin cells. Our immunocytochemical data indicate that, as it was expected by the reviewer, MCT2 (a broadly distributed lactate transporter which is abundantly expressed in neurons; Halestrap, 2013) is also clearly expressed in these two catecholaminergic cell types (Supplementary Fig. 3; page 6, lines 6-8). In agreement with this finding, we observed that in both SCG neurons and AM chromaffin cells lactate induced a fairly reversible increase in NAD(P)H autofluorescence (an effect opposed by pyruvate; Supplementary Fig. 6a,c,e). However, SCG neurons recorded with the perforated patch clamp technique did not show the depolarizing response to lactate characteristic of glomus cells, although they were reversibly depolarized by high external K⁺ (Supplementary Fig. 6b and d). We also monitored the secretory activity of chromaffin cells in slices and found that, as reported before (Muñoz-Cabello et al., 2005), they were activated by hypercapnia, however they were unresponsive to 10 mM lactate (Supplementary Fig. 6f). These data indicate that glomus cells are specific lactate sensors (page 9, lines 7-15). The differential sensitivity of glomus cells to lactate, respecting SCG neurons or AM chromaffin cells, may be a consequence of the different expression of membrane ion channels. However, it could be also due to the fact that neither SCG neurons nor AM chromaffin cells are able to generate a relevant mitochondrial ROS signal in response to lactate. Unfortunately, this question cannot be addressed experimentally now because the methodology for real-time measurement of mitochondrial (matrix or intermembrane space) ROS developed for carotid body slices (which can be maintained 48-72 hours in culture) does not work for dispersed SCG neurons or AM slices, which are maintained in healthy conditions only for a few hours.

2. Using a variety of experimental models, the authors clearly show the mechanisms of O₂ and lactate sensing by CB glomus cells are distinct, though there is some overlap in the signals involved in the transduction schema. For example, both stimuli lead to an increase in NADH and ROS generation at the mitochondrial intermembrane space (IMS), though ROS levels appear to move in opposite directions in the mitochondrial matrix.

For lactate sensing they propose the increase in NADH or NADH/NAD⁺ ratio and ROS lead to membrane depolarization and Ca²⁺-dependent secretion due to activation of a standing (Na⁺ permeant) cation conductance. Yet, a similar increase in these signals is proposed to mediate hypoxia sensing in the same cells but via inhibition of background K⁺-selective TASK1/3 ion channels. How do the authors explain the convergence of these identical signals onto two different channel types in the same cell and particularly, how is selectivity towards their respective channel type achieved when either lactate or hypoxia is present alone?

Authors: The explanation may be that lactate and hypoxia sensing are compartmentalized in glomus cells. a) Oxygen sensing: In our model, hypoxia is sensed by NADH (primarily generated at the mitochondria and shuttled to the cytosol) and ROS (generated at the mitochondrial intermembrane space) (note that ROS at the matrix decreases during hypoxia; Arias-Mayenco et al., 2018; Moreno-Dominguez et al., 2020). These mitochondria may be located near the plasma membrane in the vicinity of K⁺ channels (TASK and Kv sub-families) in what we have defined as "O₂-sensing microdomains" (Ortega-Saenz and Lopez-Barneo, 2020). For this reason, the primary action of hypoxia is to increase membrane resistance and

depolarization due to blockade of K⁺ currents. During long exposures to hypoxia the signaling molecules (NADH and ROS) may diffuse outside the microdomain thereby affecting other ion channels types. Indeed, activation of a cationic current has been reported to occur secondarily in glomus cells in response to maintained exposures to hypoxia (see Kang et al., 2014). b) Lactate sensing: NADH is directly produced in the cytosol and ROS at the mitochondrial matrix but moving out to the cytosol. The combined immediate accessibility of NADH (alone or in combination with ROS) to the cationic channels is what generates the Na⁺-dependent membrane depolarization characteristic of lactate activation. It is also possible that cytosolic acidification secondary to lactate transport contributes to activation of glomus cells. Naturally, the two mechanisms (closure of K⁺ channels and opening of cation channels) can work together to produce depolarization and Ca²⁺ influx in response to either strong stimulation or when glomus cells are simultaneously activated by hypoxia and lactate (summation or potentiation as described in Fig. 7). This “compartmentalized” model is summarized in the scheme included in Fig. 6g and in the figure legend.

3. The potential role of TRPC5 channels in mediating the lactate-induced membrane depolarization and glomus cell secretion was ruled out by the authors in experiments using TRPC5- deficient glomus cells obtained from knockout mice. However, TRPC3 channel expression is higher than TRPC5 in CB glomus cells (text refs. 30,44) and TRPC3 has been implicated in sensing of other glycolytic substrates (e.g. glucose) in hypothalamic neurons (Diabetes 2017 Feb; 66(2): 314-324). It would be interesting to know if putative ‘selective’ TRPC3 blockers (e.g. Pyr 3/10; see Cells 2018 July 7(7):83) inhibit lactate sensing in CB glomus cells.

Authors: We initially focused on TRPC5 channels because in a comparative microarray study (Gao et al., 2017) it was the TRP channel most highly expressed in CB in comparison with AM or SCG cells. In a separate gene expression study (Zhou et al., 2016) it was found that glomus cells contain abundant mRNA coding for TRPC5, TRPC3 and TRPM7, as well as beta subunit of the epithelial Na⁺ channels. Following the suggestion of the reviewer we have tested the effect of Pyr 3 (a blocker of TRPC3; Tiapko and Groschner, 2018) (Supplementary Fig. 8a,b). In addition, we have also tested the effects of FTY720 (a blocker of TRPM7; Chubanov et al., 2014) (Supplementary Fig.8c), amiloride (a blocker of epithelial Na⁺ channels; Canesa et al., 1996,) (Supplementary Fig. 8d) and the three drugs together (Supplementary Fig. 8 e,f). None of these drugs seemed to clearly inhibit glomus cell activation by lactate. Moreover, in addition to TRPC5, we have now shown that genetic ablation of either TRPC6 or TRPC5 plus TRPC6 have no effect on lactate activation of glomus cells (Supplementary Fig. 7). We have concluded that there is not a “lactate-sensing” channel but that lactate-induced NADH and ROS may redundantly modulate several subtypes of cationic channels in glomus cells (page 10, lines 1-10).

Minor comments:

1. In Abstract line 8, the authors claim “...the carotid body is essential for lactate homeostasis”. Presumably this is based largely on the observation of elevated hypoxia-induced lactatemia in CB-deficient mice, i.e. TH-HIF2a and ERT2-Cre-HIF2a mice, where the CB fails to develop and/or generates a defective HVR. However, other extra-CB cells/tissues in the peripheral and central nervous systems are affected in these HIF2a-deficient mice. Can the authors comment on whether CB denervation prevents the elevated hypoxia-induced lactatemia?

Authors: We think that the TH-Hif2a KO mice is a good model of CB “genetic ablation” because in these mice the CB is atrophied but the AM and SCG appear to be normal (Macias et al., 2018). However, we agree with the reviewer that the consequences of CB

denervation on lactate homeostasis should be now formally studied and taken into consideration. It was shown long ago (Bainton, CR J Appl Physiol Respir Environ Exerc Physiol, 44: 28-35, 1978) that ventilation induced by lactic acid is abolished by CB denervation. However, in this study the effects of lactate and acid were not studied separately. We have added a comment to the final paragraph of the Results and discussion section (page 12, lines 1-4) stressing the potential side effects of CB resection or denervation on lactate homeostasis, as this surgical procedure is used in tumor surgery (e.g. Dahan et al., 2007) and is under experimental scrutiny for the treatment of sympathetically mediated diseases (Paton et al., 2013).

2. In Fig. 3 there seems to be broad variability in the latency of the glomus cell depolarizing response to lactate. How does this compare to the latency of the hypoxia response?

Authors: The latency of the hypoxic responses studied before in our laboratory is within the range of the latency values seen in response to lactate (1-2 min). This latency is due, in part, to the dead space in our perfusion system. We have noticed that NADH signals induced by lactate tend to be faster than those induced by hypoxia. However, these differences have not been studied in detail.

3. In Extended Data Fig. 3b, should the second bin read (Lac + a-KB)?

Authors: The plot (now in Supplementary Fig. 5b) has been corrected

We thank again the reviewer for his/her constructive and helpful comments. We hope that the reviewer is satisfied with our new experimental data and more detailed explanation of the lactate signaling pathway, as well as of the differences between acute oxygen and lactate sensing.

Reviewer #3 (Remarks to the Author):

A new homeostatic function of the carotid body (CB) is proposed, which is the sensing of blood lactate leading to a upregulation of ventilation. After showing inordinate blood lactate levels in response to hypoxia in two CB-deficient mice transgenic lines, the study focuses on glomus cells in vitro. The main findings are:

- lactate induces the release of catecholamines, a phenomenon sensitive to Cd²⁺
- MCT2 is expressed
- lactate increases cytosolic Ca²⁺ and acidifies
- lactate increases UV autofluorescence, sensitive to AR-C1558585 and reverted by pyruvate
- lactate and pyruvate increase mitochondrial ROS.
- lactate depolarizes cells, which requires extracellular Na⁺
- In hyperpolarized cells, lactate triggers action potentials, reverted by pyruvate and inhibited by 2-APB
- The increase in autofluorescence and secretory response induced by lactate is insensitive to pharmacological and genetic disabling of mitochondria
- The secretory response to hypoxia is stronger in the presence of lactate.

From these results, a chain of events is proposed: lactate transport, LDH-mediated conversion into pyruvate leading to elevated NADH/NAD⁺, membrane depolarization, cation channel mediated-depolarization, activation of voltage-sensitive Ca²⁺ channels, and secretion.

This is a potentially important story. I am not aware of any homeostatic system in which

blood lactate is the regulated variable, for lactate accumulation is usually regarded as an epiphenomenon. The secretory response of glomus cells to lactate is convincing, but the characterization of the signaling mechanisms seems preliminary. The manuscript is concisely written and well organized. However, I feel that some of the data are not strong enough to support the conclusions as they are, that more experiments are needed to clarify each node of the purported signaling chain, and that some assumptions are not well supported by the literature. There are also technical issues to be addressed. Authors are invited to consider the following specific points:

Authors: We thank very much the reviewer for his detailed evaluation of our manuscript, the positive comments and specific suggestions for improvement. We have performed new experiments and taken into consideration all the comments and criticisms of the reviewer to prepare the revised version of the manuscript.

1. Is higher lactate in CB-disabled mice explained by weaker HVR?

Authors: Basal plasma levels of lactate (in normoxic conditions) are similar in wild type and in the two models of Hif2a-deficient mice used (Fig. 1d). As suggested by the reviewer, the higher lactate concentration induced by hypoxia in Hif2a-deficient mice can be explained by the lack of compensatory ventilatory response.

Otherwise, it may be attributed to general effects of HIF2 α deletion (Macias et al., 2018).

Author: Macias et al (2018) also showed that basal plasma lactate is similar in wild type and TH-Hif2a deficient mice (Macias et al., 2018, Fig. 6I). They reported that the peak level of plasma lactate reached after a bolus injection (measured 15 min after administration, Fig. 6N) is also similar in the two animal models. However, clearance of lactate is somewhat lower in TH-Hif2a KO mice than in controls (Macias et al., 2018, Fig. 6N). We think that Hif2a is required for carotid body oxygen sensing but is not primarily involved in acute lactate sensing. We have performed new experiments in conditional HIF2a-deficient mice (ERT2-HIF2a) to show that Hif2a-deficient glomus cells can be activated by lactate (Fig. 6f).

Please provide the breathing response to hypoxia of the KO animals as compared to that of controls.

Authors: The average breathing response to hypoxia of Hif2a-deficient animals in comparison with controls has already been published in previous papers from our laboratory (in Macias et al., 2018 for TH-HIF2a mice and in Moreno-Domínguez et al., 2020 for ERT2-HIF2a). Because in Macias et al., 2018 we did not show reversible plethysmographic recordings during exposure to hypoxia and hypercapnia, we thought that it was informative to present now this raw data for the TH-HIF2a model in Supplementary Fig. 1. Here, breathing responses to hypoxia of wildtype (controls) and KO mice are shown.

2. Cellular UV autofluorescence is hardly a readout of cytosolic NADH. Autofluorescence does not distinguish between NADH and NADPH, and for that reason is referred to as NAD(P)H autofluorescence, and it does not distinguish either between free and protein-bound nucleotides (which are 99% of the total). Critically, most cellular UV autofluorescence comes from the mitochondrial NADH pool and the rest is cytosolic NADPH, with an almost undetectable contribution of the very small cytosolic NADH pool (e.g. discussed by Shuttleworth et al., Neurochem. Int. 2010;). Thus, it is not possible to ascribe whole cell autofluorescence measurements to cytosolic NADH, and the current results do not demonstrate LDH-mediated oxidation of lactate. The increase may well have resulted from mitochondrial NADH accumulation induced by substrate, and/or by Ca²⁺ or acidification. Nowadays cytosolic NADH is routinely determined with precision and unambiguously using genetically-encoded sensors like Peredox, Hex, SONAR, etc.

Authors: We are aware of the limitations of cellular UV autofluorescence as a measure of NADH and, therefore, fully understand the comments of the reviewer. We are also aware of the existence of specific probes for quantitative measurement of NADH in cells. Unfortunately, these techniques have not yet been developed for CB glomus cells, which are difficult to keep viable and healthy, with reproducible responsiveness to stimuli, in primary cultures. With all necessary caveats and caution in the interpretation of the data, measurement of NAD(P)H autofluorescence has been used for years in several laboratories to obtain relevant information on the cellular processes underlying CB acute oxygen sensing (e.g. Duchen and Biscoe, 1992; Buckler and Turner, 2013; Fernandez-Aguera et al., 2015; Moreno-Dominguez et al., 2020).

To attend the request of the reviewer we have tried during the past months to develop a methodology for ratiometric NADH measurement that would work for cultured glomus cells. Initially, we focused on Peredox, as described by Hung and Yellen (2014), and obtained nice and reproducible recordings from N2A cells (a neuroblastoma cell line) bathed with solutions containing either lactate or pyruvate. Indeed, these recordings were qualitatively very similar to those that we obtained in glomus cells with UV autofluorescence. Unfortunately, Peredox did not work for primary cultures of glomus cells; the transfection of the probe seemed to damage the cells. We have also tried to get SoNar plasmids in viral vectors and, more importantly, a mouse with genetically encoded SoNar (an ideal model for our experimental purposes) recently published (Gu et al., Blood 136(5): 553, 2020). After several attempts we were able to contact one of the authors in this paper (Dr. Yi Yeng; Chinese Academy of Sciences, Shanghai) who informed us that the transgenic SoNar mice is not yet commercially available. While the SoNar mouse becomes available, we have initiated the process of signing an MTA to get SoNar plasmids from China.

While we plan to use these new tools (in particular SoNar transgenic mice) in our research on CB glomus cells in the future, we think that our current data obtained with UV autofluorescence strongly support the main points in our manuscript. Our conclusions are not based on precise quantitative measurement of free or bound NADH pools in cytosol but on the recording of fast “changes in fluorescence” in response to lactate/pyruvate stimulation/inhibition. In addition, UV autofluorescence measurements are complementary of several other sets of data obtained with different recording techniques (amperometry, patch clamp) and pharmacological analyses.

Naturally, for the correct interpretation of our UV fluorescent measurements we have taken into consideration the potential confounding factors mentioned by the reviewer:

The increase may well have resulted from mitochondrial NADH accumulation induced by substrate, and/or by Ca²⁺ or acidification.

Authors: a) The fast changes in UV fluorescence elicited by lactate cannot be explained by mitochondrial NADH accumulation induced by substrate because pyruvate (which should be more swiftly metabolized by mitochondria than lactate) systematically opposed the effect of lactate (Fig. 4e). We have also shown that, similar to pyruvate, α -ketobutyrate (which is converted to non-metabolizable α -hydroxybutyrate) also opposes the effect of lactate (Supplementary Fig. 5a,b). Moreover, fast changes in UV fluorescence elicited by lactate (10 mM) are recorded in the presence of rotenone (Fig. 6a) or in TH-NDUFS2 mice (Fig. 6b), conditions in which mitochondria NADH metabolism is strongly inhibited.

b) The fast changes in UV fluorescence induced by lactate cannot be the result of an increase in cytosolic Ca²⁺ because pyruvate produces a marked decrease in the UV autofluorescence signal (Fig. 4e; Arias-Mayenco et al., 2018, Fig. 4b). At very high concentrations (5 mM; ~50 times the normal plasma concentration) pyruvate, in the absence of lactate, induces Ca²⁺-dependent exocytosis in glomus cells (Supplementary Fig. 5f). Moreover, lactate can induce fast changes in UV autofluorescence in SCG neurons although they are not depolarized (Supplementary Fig. 6a-d). Similarly, lactate induces fast changes in

UV autofluorescence in AM chromaffin cells without inducing an increase in Ca^{2+} -dependent exocytosis (Supplementary Fig. 6e,f).

c) The fast changes in UV fluorescence induced by lactate cannot be explained by an intracellular acidification because pyruvate and α -ketobutyrate, which are also co-transported with protons, have an effect opposed to lactate (Fig. 4e; Supplementary Fig. 5a).

3. Even if a cytosolic NADH were verified with a genetically-encoded sensor, demonstration of a causal connection to cell depolarization, Ca^{2+} rise and catecholamine secretion needs more experiments. For example independent manipulation of cytosolic NADH by addition of extracellular NADH. Against a major role for cytosolic NADH, pyruvate, which reduces cytosolic NADH, also induced a robust secretory response in glomus cells (Extended Data Fig. 4B).

Author: The lactate -induced increase in NADH is associated to cell depolarization, Ca^{2+} rise and catecholamine secretion because all experimental tests that inhibit this lactate-induced signal (pyruvate, α -ketobutyrate) also inhibit cell depolarization and secretion.

Following the reviewer suggestion we have performed experiments to test the effect of extracellular NADH (5-50 micromol) on glomus cells (reported resting values for NADH in human plasma are between 0.05 and 1.2 micromol; Shinghal and Zhang, The FASEB J, 20: A1357, 2006). We did not see any appreciable effect of NADH on intact cells. In some cells recorded with perforated patches, NADH seemed to alter seal stability.

As mentioned before, pyruvate systematically inhibits all the processes induced by lactate (NADH signal, cell depolarization and secretion) (Figs. 4e,f,g and Fig. 5e). We have shown that in the absence of lactate, pyruvate at very high concentration (5 mM, ~50 times the normal plasma concentration) induces a slower secretory response in the cells (Supplementary Fig. 5f). This action of pyruvate can be explained by the combined action of intracellular acidification (due to pyruvate co-transport with protons) and mitochondrial ROS (page 7, lines 1-3 from the bottom).

4. Similar effects of lactate and pyruvate suggest that the secretory response may relate to mitochondrial fuelling.

Author: We think that our data strongly suggest that the acute activation of glomus cells by lactate is associated to an increase in cytosolic NADH, although other factors (mitochondrial ROS and acidification secondary to lactate uptake) also contribute to the secretory response. In the absence of lactate, pyruvate decreases cytosolic NADH levels but at the same time lowers pH and increases ROS formation thereby inducing a secretory response in the cells.

Even in the presence of glucose, cytosolic pyruvate is often near zero if there is no pyruvate in the medium (e.g. San Martin et al., PLoS One 2014; Contreras-Baeza et al., JBC 2019). This "pre-starvation" hypothesis may explain why both lactate and pyruvate evoked mitochondrial ROS responses and why α -ketobutyrate prevented the effect of lactate. A confounding effect of pyruvate shortage could be tested by trying lactate-stimulation in physiological pyruvate (100 μM).

Author: We have tested that 100 μM pyruvate inhibits responsiveness to lactate of glomus cells (Supplementary Fig. 5c, d). Pyruvate is transported with high affinity by MCT2 and once in the cytosol rapidly changes the lactate/pyruvate equilibrium and consumes NADH.

Secondarily, pyruvate may be transported to mitochondria to produce NADH in the Krebs cycle. However, 100 μM pyruvate is a non-physiological condition as the ratio lactate/pyruvate in plasma is 10 or higher (Psychogios et al., 2011). We have done new experiments in cells maintained in physiological extracellular levels of lactate (1.4 mM) and pyruvate (0.1 mM). In these conditions an increase in lactate (simulating what would occur in

an animal exposed to systemic hypoxia) resulted in a powerful activation of glomus cells (Fig. 4h).

Another effect common to lactate and pyruvate is cytosolic acidification, which is bound to affect matrix pH and mitochondrial metabolism (Azarias et al. J. Neurosci. 2011) .

Author: Yes, as it was indicated above, cytosolic acidification may contribute to the secretory responses elicited by lactate and by pyruvate (in the absence of lactate).

5. The Ca^{2+} response warrants a deeper characterization. Sensitivity to Cd^{2+} is not enough to decrete extracellular Ca^{2+} involvement, as Cd^{2+} inhibits all kinds of channels and transporters. Could acidification contribute to the Ca^{2+} rise, perhaps by engagement of NHE and NCX? That might help to explain the Na^{+} sensitivity of the effect and why pyruvate also induced a secretory response. Is there a contribution of Ca^{2+} stores?

Author: We normally use Cd^{2+} to non-selectively block Ca^{2+} entry in glomus cells because they have several subtypes of high-voltage activated Ca^{2+} channels, although the proportion of L-type channels (the most abundant) versus other channel types (P, Q, or N types) changes among animal species. We have shown before that 0.2 mM Cd^{2+} completely blocks high-threshold Ca^{2+} currents in glomus cells (Urena et al., 1994). Following the suggestions of the reviewer we have now tested the effect of nifedipine (a specific L-type channel blocker). As expected, nifedipine strongly inhibits glomus cell secretory response to lactate (Fig. 2i, j).

We have no evidence of Ca^{2+} release from stores playing any relevant role in responsiveness of glomus cells to lactate.

6. There is variability in the onset of the lactate effects, which makes it hard to visualize a causal chain. According to Extended Data Fig. 2, Ca^{2+} rises within 1 minute of lactate exposure but pH starts to drop afterwards. This delay does not support the suggested chain of events. If this variability is not due to perfusion issues, please provide simultaneous measurements of the critical variables.

Authors: Most of the latency in the initiation of the responses to lactate (and to other stimuli) is due to the dead space in our perfusion system. This can change several seconds in different experiments. Honestly, our perfusion system (based on gravity, syringes, needles, PE tubing, open recording chamber with suction, etc) is too simple to allow for any formal discussion on latencies of the different stimuli.

7. MCTs. Immunocytochemistry of MCTs is notorious for false positive and negatives as commercial antibodies are suspect and epitopes are often masked. It is not clear what is the point of showing MCT2 expression, without reference to MCT1, which is also blocked by AR-C155858.

Authors: Following the suggestion of this reviewer (and the two other reviewers) we have performed several new experiments to characterize in further detail the expression of the most relevant MCTs (MCT1, 2 and 4; Halestrap, 2013) in carotid body glomus (O_2 -sensitive) and type II cells. Expression of MCT2 in glomus cells is clearly confirmed. These cells also express MCT4 but do not seem to express MCT1 (Fig. 3b, Supplementary Figs. 2 and 4).

Moreover, most of the staining seems to be cytoplasmic (Mitochondria perhaps?). More informative would be MCT 1/2 immunoblots of plasma membranes.

Authors: We think that the quality and definition of the immunocytochemical analysis has improved. The mouse CB is very small (about 0.5 mm in diameter) and therefore the amount of tissue available is too small to allow purification of plasma membranes. In any instance, it is well established that MCT2 is expressed in the membrane of neurons and other cell types.

8. Introduction. The notion that lactate is the final product of glycolysis is correct for cells that export lactate, like white muscle fibers and brain astrocytes, but not for those that consume lactate, like red muscle fibers and neurons (discussed by Barros et al., Neurochem. Res. 2020). Authors should also be aware that the intracellular lactate shuttle is far from “well established” a concept, rather the opposite (Glancy et al., J. Physiol. 2020).

Author: We agree with the comment of the reviewer. The first paragraph of the Introduction has been rewritten to clarify the point raised by the reviewer (page 3, lines 1-8). The references Barros et al., 2020 and Glancy et al., 2020 have been added.

Minor

9. Citations. Reference number 3 is not cited, whereas number 5 relates to tumors, not to healthy tissues. Reference number 6 is a comment on number 3. Please check the paper throughout.

Authors: References have been checked. References to tumor tissues (Faubert et al., 2017; Martinez-Reyes and Chandel, 2017) have been removed. Several new references (in red in the reference list) have been added.

10. Pg. 4, line 8. “did evoked an ventilatory”.

Authors: Corrected

11. Pg. 5, line 5. “directly”. The pH technique is indirect, direct monitoring is done with lactate or pyruvate sensors.

Authors: “Directly” has been removed.

12. Pg 7, line 7. “signaling. molecules”

Authors: Corrected.

We thank again the reviewer for his constructive and helpful comments. We hope that he is satisfied with the new experimental data and more detailed explanation of the lactate signaling pathway included in the revised version of our manuscript.

Reviewer comments, second round: –

Reviewer #1 (Remarks to the Author):

The authors provided a comprehensive response to my comments and made significant modifications to the manuscript. The manuscript was very interesting in the first draft and now it is further improved and definitely represents a major contribution to the field. I have several comments:

- 1) Page 5, 1st paragraph. The statement 'These data indicate that although CB stimulation induces sympathetic activation, this process is not involved...' This conclusion has to be substantiated. The authors probably meant to say 'hypercapnea induces sympathetic activation via CB, but it is not accompanied by lactatemia'? If I am correct, they may consider to include a panel or a figure showing catecholamine release in response to hypercapnea compared to hypoxia and lactate (in Fig. 2B, for instance?).
- 2) Page 7, 2nd paragraph. Is pyruvate co-transported with H⁺ by MCT1 as well?
- 3) Page 9, 1st paragraph. What explains why SCG neurons are insensitive to lactate unlike CB glomus cells?

A minor comment:

Should abbreviation MCI - mitochondrial complex I - be explained?

Reviewer #2 (Remarks to the Author):

The authors have added substantive new data that have greatly enhanced the impact of the manuscript. The new experiments showing that SCG neurons and adrenal chromaffin cells are lactate insensitive are welcomed additions, supporting the posit that glomus cells are specialized lactate sensors. Also, the new pharmacological and genetic studies on the potential roles of the various TRP channels in lactate sensing by glomus cells were informative, even though the findings were mostly negative. My remaining comments for the authors are relatively minor (see below).

- In contrast to the authors' findings, a recent paper claims that, unlike mice, rat carotid body and glomus cells are insensitive to lactate (Spiller PF et al., *Resp. Physiol. & Neurobiol.* 285, 2021, 103593). Though recording conditions and data interpretation are questionable in the latter paper, the present authors may nevertheless wish to include and comment on that paper.

-p7, line 6 from bottom- replace 'where' by 'were'

-p9, line 4 from bottom- change to ... mRNA is more highly expressed....

-p33 Fig. 5 line 7. To remove ambiguity and for clarity, I suggest sentence be re-arranged as follows:..were: 9.5 +/- 1 mV (n =17 cells from 16 mice.....) at RP; 31.2 +/- 4 mV (.....) at ~ -70 mV. This wording avoids the misinterpretation that the mean RP was 9.5 mV.

-p37, Fig 7a- change labeling from 'lactato' to lactate.

Supplementary information;

-p6, Supp Fig. 5 legend, line 2- spelling .. α -ketobutyrate

Colin Nurse

Reviewer #3 (Remarks to the Author):

Authors have made a substantial effort to address the reviewers' comments, involving clarifications and numerous new experiments. It is understood that cytosolic NADH measurements are not feasible at this point, an aspect that will be left for the future. Many thanks for the hard work and congratulations for an important article!

I have only one comment that authors may care to consider for the final version of the manuscript:

Revised section in Pg. 6, regarding MCT4.

MCT4 was long thought to be a low lactate affinity transporter without selectivity for pyruvate, a notion based on indirect pH measurements (ref #37). Recently, our group characterized MCT4 more directly using a genetically-encoded lactate sensor (Contreras-Baeza et al., JBC 294(52):20135-20147, 2019, doi: 10.1074/jbc.RA119.009093.2019). We concluded that pH measurements were biased, that MCT4 has high affinity for lactate (Km of 1 mM), and that MCT4 does transport pyruvate, albeit with lower affinity (Km of 4 mM).

As shown by a numerical simulations included in the same article (Fig, 8), these features of MCT4 permit cells to export lactate in high lactate environments. Of significance for the present article, MCT2 is poised to import lactate.

RESPONSE TO REVIEWERS' COMMENTS

Reviewer #1 (Remarks to the Author):

The authors provided a comprehensive response to my comments and made significant modifications to the manuscript. The manuscript was very interesting in the first draft and now it is further improved and definitely represents a major contribution to the field. I have several comments:

Authors: We very much appreciate the constructive comments of the reviewer and his/her contribution to the improvement of the manuscript.

1) Page 5, 1st paragraph. The statement 'These data indicate that although CB stimulation induces sympathetic activation, this process is not involved...' This conclusion has to be substantiated. The authors probably meant to say 'hypercapnea induces sympathetic activation via CB, but it is not accompanied by lactatemia'? If I am correct, they may consider to include a panel or a figure showing catecholamine release in response to hypercapnea compared to hypoxia and lactate (in Fig. 2B, for instance?).

Authors: Yes. We have modified the wording of the sentence as indicated by the reviewer (page 5, first paragraph). Responses of glomus cells to hypercapnia compared to hypoxia have been published before by our group. Reference 27 (a recent review) is indicated.

2) Page 7, 2nd paragraph. Is pyruvate co-transported with H⁺ by MCT1 as well?

Authors: Yes. A short comment on this is included (page 7, lines 6,7 of second paragraph).

3) Page 9, 1st paragraph. What explains why SCG neurons are insensitive to lactate unlike CB glomus cells?

Author: We have included the following paragraph (page 10, lines 2-6):

"The differential sensitivity of glomus cells to lactate, respecting SCG neurons or AM chromaffin cells, may be a consequence of the different expression of membrane ion channels. However, it could be also due to the fact that neither SCG neurons nor AM chromaffin cells are able to generate a relevant mitochondrial ROS signal in response to lactate".

A minor comment:

Should abbreviation MCI - mitochondrial complex I - be explained?

Author: MCI is defined on page 11, line 8.

Reviewer #2 (Remarks to the Author):

The authors have added substantive new data that have greatly enhanced the impact of the manuscript. The new experiments showing that SCG neurons and adrenal chromaffin cells are lactate insensitive are welcomed additions, supporting the posit that glomus cells are specialized lactate sensors. Also, the new pharmacological and genetic studies on the potential roles of the various TRP channels in lactate sensing by glomus cells were informative, even though the findings were mostly negative. My remaining comments for the authors are relatively minor (see below).

Authors: We very much appreciate the constructive comments of the reviewer and his contribution to the improvement of the manuscript.

- In contrast to the authors' findings, a recent paper claims that, unlike mice, rat carotid body and glomus cells are insensitive to lactate (Spiller PF et al., Resp. Physiol. & Neurobiol. 285, 2021, 103593). Though recording conditions and data interpretation are questionable in the latter paper, the present authors may nevertheless wish to include and comment on that paper.

Authors: We thank the reviewer for bringing this point to the discussion. In response to the recent publication by Spiller et al. (which we noticed once our manuscript had been submitted) we have performed experiments in rat CB cells. Our data clearly show that similar to the findings in the mouse CB, lactate activates (increase in cytosolic Ca²⁺ and catecholamine release) rat glomus cells. We have added a Supplementary figure (Fig. S2) and a sentence explaining our results (page 5 last 5 lines and page 6 first two lines). Reference Spiller et al., has been included.

-p7, line 6 from bottom- replace 'where' by 'were'

Authors: Done as indicated by the reviewer (page 8, line 5).

-p9, line 4 from bottom- change to ... mRNA is more highly expressed....

Authors: Done as indicated by the reviewer (page 10, line 6 second paragraph)

-p33 Fig. 5 line 7. To remove ambiguity and for clarity, I suggest sentence be re-arranged as follows:..were: 9.5 +/- 1 mV (n =17 cells from 16 mice.....) at RP; 31.2 +/- 4 mV (.....) at ~ -70 mV. This wording avoids the misinterpretation that the mean RP was 9.5 mV.

Authors: Done as indicated by the reviewer (figure legend)

-p37, Fig 7a- change labeling from 'lactato' to lactate.

Authors: Done as indicated by the reviewer

Supplementary information;

-p6, Supp Fig. 5 legend, line 2- spelling .. α -ketobutyrate

Authors: Done as indicated by the reviewer (now Supp. Fig. 6 legend)

Colin Nurse

Reviewer #3 (Remarks to the Author):

Authors have made a substantial effort to address the reviewers' comments, involving clarifications and numerous new experiments. It is understood that cytosolic NADH measurements are not feasible at this point, an aspect that will be left for the future. Many thanks for the hard work and congratulations for an important article!

I have only one comment that authors may care to consider for the final version of the manuscript:

Authors: We very much appreciate the constructive comments of the reviewer and his contribution to the improvement of the manuscript. We also want to apologize for having missed the relevant study by Contreras-Baeza et al. (see below).

Revised section in Pg. 6, regarding MCT4.

MCT4 was long thought to be a low lactate affinity transporter without selectivity for pyruvate, a notion based on indirect pH measurements (ref #37). Recently, our group characterized MCT4 more directly using a genetically-encoded lactate sensor (Contreras-Baeza et al., JBC 294(52):20135-20147, 2019, doi: 10.1074/jbc.RA119.009093.2019). We concluded that pH measurements were biased, that MCT4 has high affinity for lactate (K_m of 1 mM), and that MCT4 does transport pyruvate, albeit with lower affinity (K_m of 4 mM).

As shown by a numerical simulations included in the same article (Fig, 8), these features of MCT4 permit cells to export lactate in high lactate environments. Of significance for the present article, MCT2 is poised to import lactate.

Authors: We have done some changes in the text to include the new findings regarding the relatively high affinity of MCT4 for lactate and pyruvate (page 6, last 9 lines; page 7 first line). The reference Contreras-Baeza et al. has been added.